# Predictors of multiple sclerosis progression: A systematic review of conventional magnetic resonance imaging studies

**Nima Broomand Lomer**[1]*, **Kamal AmirAshjei Asalemi**[1], **Alia Saberi**[2], **Kasra Sarlak**[1]

**1** Faculty of Medicine, Guilan University of Medical Sciences, Rasht, Iran, **2** Department of Neurology, Poursina Hospital, Faculty of Medicine, Guilan University of Medical Sciences, Rasht, Iran

* nima.broomand@gmail.com

**Data Availability Statement:** All relevant data are within the manuscript and its Supporting information files.

## Abstract

### Introduction

Multiple Sclerosis (MS) is a chronic neurodegenerative disorder that affects the central nervous system (CNS) and results in progressive clinical disability and cognitive decline. Currently, there are no specific imaging parameters available for the prediction of longitudinal disability in MS patients. Magnetic resonance imaging (MRI) has linked imaging anomalies to clinical and cognitive deficits in MS. In this study, we aimed to evaluate the effectiveness of MRI in predicting disability, clinical progression, and cognitive decline in MS.

### Methods

In this study, according to PRISMA guidelines, we comprehensively searched the Web of Science, PubMed, and Embase databases to identify pertinent articles that employed conventional MRI in the context of Relapsing-Remitting and progressive forms of MS. Following a rigorous screening process, studies that met the predefined inclusion criteria were selected for data extraction and evaluated for potential sources of bias.

### Results

A total of 3028 records were retrieved from database searching. After a rigorous screening, 53 records met the criteria and were included in this study. Lesions and alterations in CNS structures like white matter, gray matter, corpus callosum, thalamus, and spinal cord, may be used to anticipate disability progression. Several prognostic factors associated with the progression of MS, including presence of cortical lesions, changes in gray matter volume, whole brain atrophy, the corpus callosum index, alterations in thalamic volume, and lesions or alterations in cross-sectional area of the spinal cord. For cognitive impairment in MS patients, reliable predictors include cortical gray matter volume, brain atrophy, lesion characteristics (T2-lesion load, temporal, frontal, and cerebellar lesions), white matter lesion volume, thalamic volume, and corpus callosum density.

**Funding:** The author(s) received no specific funding for this work.

**Competing interests:** The authors have declared that no competing interests exist.

## Conclusion

This study indicates that MRI can be used to predict the cognitive decline, disability progression, and disease progression in MS patients over time.

## Introduction

Multiple Sclerosis (MS) is a chronic autoimmune disease that affects the central nervous system (CNS) and leads to demyelination, axonal loss, and neurodegeneration. The disease is caused by a complex interaction of environmental and genetic factors that are not yet fully understood [1, 2]. MS presents with a wide range of symptoms including sensory disturbances, walking difficulties, vision problems, intestinal and urinary dysfunction, cognitive and emotional impairment, dizziness, vertigo, sexual problems, speech difficulties, seizures, and headaches [3, 4]. MS is classified into four subgroups based on phenotype: clinically isolated syndrome (CIS), relapsing-remitting MS (RRMS), secondary-progressive MS (SPMS), and primary-progressive MS (PPMS). RRMS is the most common form of the disease, affecting approximately 85% of patients at presentation. It is characterized by acute exacerbations followed by clinically stable periods [5]. PPMS, on the other hand, presents with a slowly progressive reduction in neurological function from the start without clinical relapses [6, 7]. Naturally, RRMS tends to convert to SPMS which is an irreversible gradual disability progression [8]. In the past, nearly 10% of RRMS patients progressed to SPMS in a 5-year period, 25% in 10 years, and 75% in 30 years. However, with the advent of more treatment options and early diagnosis, the risk of SPMS conversion has decreased to about 2%, 9%, and 27% in a 10-year, 15-year, and 20-year period, respectively [9–11]. In addition to physical disability, impairment of cognitive function is also a common manifestation of MS. Neuropsychological abnormalities are observed in 40–70% of MS patients, and cognitive impairment is a predictor of disease progression [12]. MS in cognitively impaired patients is more likely to progress in upcoming years [13]. The most common cognitive impairments in MS include reduced speed of information processing and working memory, which can disrupt data retention ability and short-term memory [14–17]. Unfortunately, the underlying mechanisms of cognitive impairment in MS are not yet fully understood [18, 19].

Magnetic resonance imaging (MRI) plays a pivotal role in the detection, prognosis, and evaluation of disease activity in MS [20–23]. Focal lesions, atrophies, and normal appearing tissue damages are among the MS pathologies that can be detected using MRI [20]. White matter lesions and deep gray matter atrophy typically arise in the early stages of the disease, while cortical atrophy and demyelination emerge in later stages [24–27].

Features of MS lesions in the brain or spinal cord, including the presence of lesions or changes in the size of certain CNS structures such as the thalamus, corpus callosum, cerebellum, limbic system, and spinal cord are not addressed in the latest version of McDonald Criteria (2017) [23] or in recent guidelines for determining disease progression or deciding for escalation or change of treatment in MS disease. Considering this, here we aimed to evaluate the potential of conventional MRI markers in predicting clinical disability, disease progression, and cognitive decline in MS patients.

## Methods and materials

### Eligibility criteria

We included studies with the following criteria: [1] Definite diagnosis of MS based on the revised McDonald's criteria of 2017 [2, 23], Applied conventional MRI, and [3] Focused on

evaluating the progression of disability or cognitive decline in MS patients. To ensure the quality of the data, we excluded various types of publications, including review articles, animal studies, letters and commentaries, case reports, case series, book chapters, conference abstracts, and non-English studies. Furthermore, the study only included research conducted among patients with RRMS or progressive forms of the disease, while studies conducted among patients with CIS were excluded. We excluded studies with the usage of AI (Deep learning and Machine learning methods) in the prediction of course of disease.

## Search strategy

We conducted this systematic review according to the guideline of preferred reporting items for Systematic reviews and Meta-Analysis (PRISMA) [28]. Search was performed in PubMed, Embase and Web of Science databases from 2010 until July 2023 to identify the relevant studies using the keywords below:

> *("progressive multiple sclerosis" OR "Multiple Sclerosis, Chronic Progressive"[Mesh] OR "progressive MS" OR "primary progressive multiple sclerosis" OR "secondary progressive multiple sclerosis" OR "primary progressive MS" OR "secondary progressive MS" OR PPMS OR SPMS) AND ("relapsing remitting multiple sclerosis" OR "relapsing-remitting multiple sclerosis" OR "relapsing-remitting MS" OR "relapsing/remitting multiple sclerosis" OR "relapsing/remitting MS" OR "Multiple Sclerosis, Relapsing-Remitting"[Mesh] OR "relapsing-remitting MS" OR"relapse-onset MS" OR "relapse-onset multiple sclerosis" OR RRMS) AND (MRI OR "magnetic resonance imaging" OR "magnetic resonance imaging"[MeSH] OR imaging) AND (2010:2023 [pdat])*

We made a slight adjustment to our search strategy to integrate with two other databases. Initially, there were no restrictions on the type of studies, their location, or language. We screened and extracted data from all studies conducted from 2010 to July 2023 using EndNote software [29]. Flow diagram of the database searching and study selection according to the PRISMA guideline is presented below in Fig 1.

## Screening and data extraction

This stage was conducted in three distinct phases by two independent authors, namely K.A.A and K.S. In the first phase, the titles and abstracts of the records were carefully screened to determine their initial eligibility for inclusion in the study. In the event of any discrepancies, the third and fourth authors, N.B.L and A.S, were consulted to resolve the issue by consensus. In the second phase, the full text of the selected records was retrieved. Only those articles that specifically studied MRI markers in relation to disability progression or cognitive decline in RRMS or progressive forms of MS were included. In the third and final phase, relevant data was extracted and recorded in a data collection table, which included important information such as the demographic features of the study participants (year of study, number and studied groups of participants, mean age, and disease duration), the imaging methodology used (field strengths in Tesla and studied parameter), and the correlations of MRI markers with disability progression and cognitive decline.

## Data items

In this review, we aimed to assess any correlations of MRI markers with disability progression and cognitive decline in RRMS and progressive forms of MS. These terms are defined as follows:

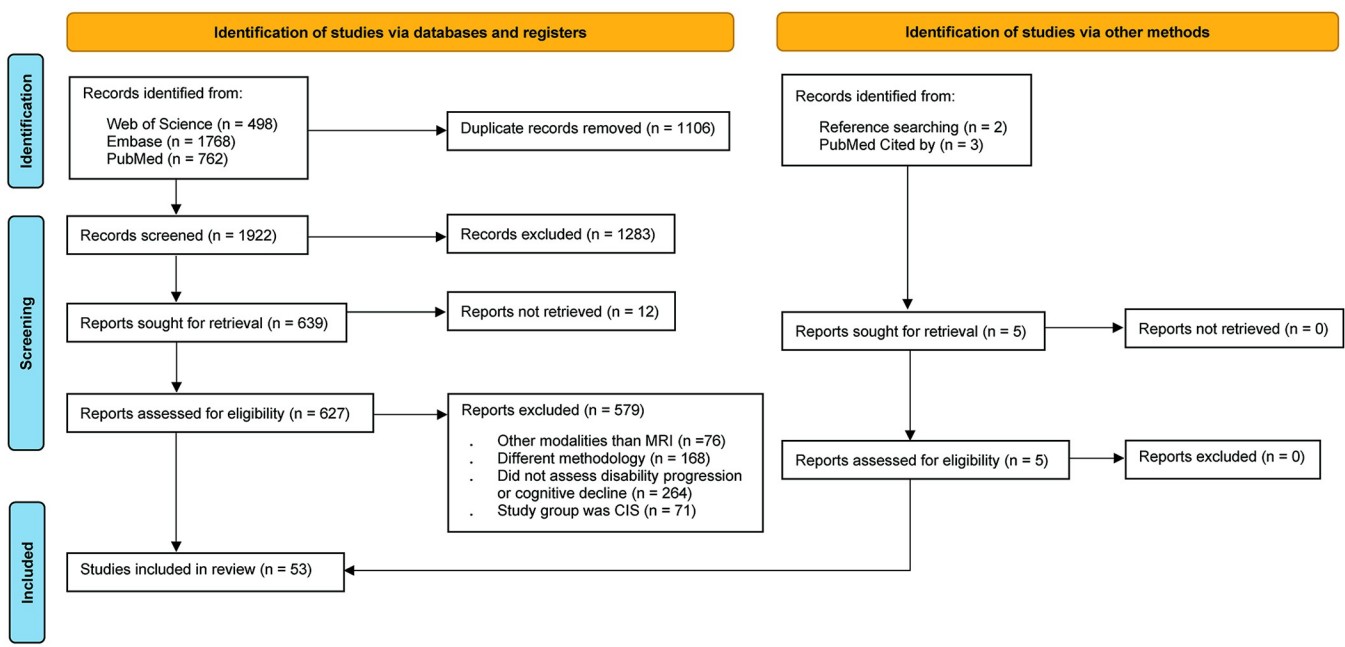

**Fig 1. PRISMA flow diagram of database searching and study selection.**

**Disability:** Disability progression in MS is a broad term referring to the worsening of physical, cognitive and emotional symptoms during the disease course. We mainly aimed at the physical disability mostly measured by Expanded Disability Status Scale (EDSS) among included studies. Depending on the affected area of CNS, physical disability progression can manifest with different symptoms including muscle weakness, balance and coordination problems, fatigue, tremors and difficulty walking. In addition, Timed 25-Foot Walk (T25FWT) and 9-Hole Peg Test (9HPT) can be utilized to assess disability outcomes. The T25FWT assesses an individual's time to walk 25 feet as swiftly as possible while ensuring safety. Prolonged completion times indicate increased disability levels. Meanwhile, the 9HPT evaluates arm and hand functionality, employing a small container with nine holes and pegs. Participants are instructed to place and remove the pegs from the holes individually as rapidly as possible. Longer completion times signify higher disability.

**Progression:** The progression of MS refers to how the disease evolves and advances over time. CIS and RRMS phenotypes tend to progress and convert to the progressive phenotypes of MS.

**Cognition:** Cognitive decline, a representative of disability progression in MS, refers to the progressive deterioration of cognitive functions, including memory, attention, information processing speed, executive functions, and problem-solving abilities. It can significantly impact daily functioning, work performance, and overall quality of life. Two commonly used cognitive function assessment tools in studies are the Symbol Digit Modalities Test (SDMT) and the Paced Auditory Serial Addition Test (PASAT). The SDMT evaluates processing speed and attention by matching symbols with numbers within a time limit, while the PASAT measures processing speed, flexibility, and working memory by requiring participants to add orally presented numbers in sequence.

## Risk of bias assessment

In accordance with the PRISMA guidelines [28], the quality assessment of the studies included in this systematic review was conducted using the Joanna Briggs Institute Critical Appraisal tools (JBI) specifically designed for cross-sectional studies [30]. Two independent authors (N.B.L and K.S) conducted the assessment by answering 11 questions that evaluate different domains of the studies to ascertain their potential risk of bias. The questions could be answered with 'yes', 'no' or 'unclear'. Any discrepancies between the two reviewers were discussed and resolved by achieving a consensus. The risk of bias for each individual study was determined based on the following criteria: low risk of bias if 70% of answers scored yes, moderate risk if 50 to 69% questions scored yes and high risk of bias if yes scores were below 49% [31].

## Results

A total of 3028 articles were identified after conducting a thorough search of the database, including 762 articles from PubMed, 498 from Web of Science, and 1768 from Embase. Following the elimination of duplicates, 1922 articles remained for title and abstract screening. Subsequently, 639 articles were retrieved for a full text analysis, out of which 12 records were not found. During the comprehensive full text screening, 579 articles were excluded for not meeting the inclusion criteria. An additional five studies were found through other sources and met the inclusion criteria, leading to a total of 53 studies being included in this review.

Correlating MRI markers in cortical lesions and gray matter alterations of MS patients were assessed in fifteen studies, spinal cord alterations in twelve studies, corpus callosum alterations in three studies, cerebellum alterations in six studies, thalamus alterations in eleven studies, limbic system alterations in two studies, lesion atrophy in two studies, whole brain and white matter lesion volume in nineteen studies.

## MRI markers predicting the disability progression

Abnormalities in the gray matter, whether deep or cortical, including atrophy or lesion in the cortex, can predict the progression of disability among patients with MS. Several studies have shown a strong correlation between cortical lesion and EDSS score [32–38] with cortical lesion volume being a predictor of neurologic disability progression during follow-up [35]. Gray matter atrophy has also been identified as a predictor of higher EDSS scores [34, 39, 40]. In addition, the ratio of gray matter (GM) to normal appearing white matter (NAWM) in recently diagnosed RRMS patients can predict disability progression [41]. The deep gray matter has been found to be a predictor of time-to-EDSS progression [42].

In white matter (WM), EDSS score was significantly correlated with WM lesion volume, central atrophy, lesion probability in the periventricular WM at the left frontal horn and around the posterior horns and with whole-brain volume particularly with widths of third and lateral ventricle [32, 37, 43–45]. The presence of confluent lesions (in RRMS), higher number of T2 lesions, lower baseline T2-lesion volume (T2LV), lower normalized brain volume (NBV), higher percentage brain volume change (PBVC) between year 2 and baseline and presence of $\geq$ 4 slowly expanding lesions (SELs) were defined as prognostic factors for EDSS worsening and disability progression [46–48, 83].

In the corpus callosum, some indices, including corpus callosum index (CCI) and corpus callosum lesion volume (CCLV), which indicate corpus callosum (CC) damage, were associated with disability progression and EDSS change [49, 50].

Thalamic volume change especially in the anterior, ventral anterior, ventral lateral and pulvinar nuclei inversely correlated with EDSS [50–53]. Furthermore, the EDSS was negatively associated with thalamic iron [54].

Spinal cord changes such as atrophy (GM and WM) or lesions were indicators of disability and worsening EDSS. Some studies suggested that smaller cervical cross-sectional area (CS-SCA), especially CSA-C2, loss of spinal cord volume (SCV), baseline annualized percentage upper cervical cord cross-sectional area change (aUCCA), and the number of spinal cord segments affected by T2-lesions are all predictive factors for disability [34, 40, 55–60].

In all CNS structures, atrophied lesion volume was significantly associated with disability progression [61].

## MRI markers predicting the progression of disease

Cortical lesions and gray matter volume are two most significant determinants of a progressive disease [34, 35]. Cortical lesions are more prevalent in SPMS subjects compared to RRMS subjects [35] and higher baseline cortical lesions predicted conversion to SPMS [62]. Temporal gray matter atrophy is faster in SPMS than RRMS [42]. The GM/NAWM ratio is a predictor of SPMS conversion in recently diagnosed RRMS patients, implicating that GM and NAWM are influenced differently regarding disease development since early stages of MS [41].

Some MS lesion characteristics and also atrophy of brain are among other key markers and predictors of MS progression. Notably, iron rims serve as a representation of the chronic active nature of MS lesions, indicating a more severe and damaging form of the disease [63]. In a study, the only longitudinal MRI marker that was capable of distinguishing patients who deteriorated gradually from those who remained stable was brain atrophied T2-LV [64]. In another 9.1-year longitudinal study, the number and volume of T2 hyperintense lesions and lower NBV were significantly associated with conversion to SPMS [47]. Higher annualized percentage ventricular volume change (aPVVC) during the first 2 years was observed in patients with progressive disease compared to patients with no progression [65]. Central atrophy was associated with disease progression over 5.5 years in early RRMS [45]. One study showed that significant discriminative MRI atrophy measurements in RRMS vs SPMS are as followed: Index of frontal atrophy, Index of EVANS, Huckman Index, Bicaudatus Index and Width of third ventricle. For differentiating RRMS from SPMS; Remission-Progression Index formula can be used [66]: Remission-Progression Index = (RAVLT 1–5 SUM + DSST)/Huckman Index.

The corpus callosum index is an important prognostic factor for the progression of MS. It has been observed that individuals diagnosed with SPMS exhibit lower levels of CCI at the time of diagnosis, while also experiencing a greater decline in annual CCI compared to those with RRMS [67].

Deeper nuclei impairment, higher thalamic lesion volume and higher thalamic volume reduction was seen in SPMS compared with the RRMS group [38, 51, 68]. Baseline volume and the rate of annual volume loss of the ventral lateral nucleus were significant predicting factors of disease progression [53].

Spinal cord abnormalities including atrophy and lesion and gadolinium-enhancement at disease onset and during disease are also predictors of MS progression and conversion to progressive forms [69]. SCV loss particularly cervical GM atrophy is a predicting factor for progression. Although, cervical CS-SCA, especially cross-sectional area of C2 (CSA-C2) is significantly smaller in PMS compared to RRMS, but thoracic SCAs are not significantly different between types of MS [55–57, 60]. Reduction of UCCA over 24 months is seen on all MS types and is higher in SPMS [59]. Patients who develop SPMS exhibit accelerated cord atrophy

rates before conversion and decelerated rates after conversion [70]. Clusters of cord atrophy are mainly found in the lateral and posterior cord segments [71].

Atrophy of the cerebellum, especially cerebellar posterior superior lobe atrophy was higher in SPMS compared to RRMS [72–74]. Significantly higher volumes and numbers of cerebellar cortical lesions were found in SPMS and PPMS compared to RRMS and CIS [75, 76]. Although these changes manifest during the progression of the disease and may not be immediately apparent at the onset of the condition, hence they may not be regarded as reliable predictors of disease progression.

## MRI markers predicting the cognitive decline

Cortical gray matter volume was an MRI predictor of cognitive decline [77]. But cortical lesion (CL) volume and CL load were not significant predictors of neuropsychological outcomes, and were only associated with impairing the more challenging cognitive tests such as Trail Making Test (TMT-B) [32, 33].

Brain atrophy was correlated with verbal memory impairment and other neurocognitive symptoms. Third ventricle width and bicaudatus ratio correlated mostly with the performed cognitive tests particularly Symbol Digit Modalities Test (SDMT) [66]. In RRMS, atrophy of WM was correlated with verbal memory performances [78].

Some MS lesion characteristics were predictors of cognitive impairment. Atrophied T2-LV among PMS patients, was related to follow-up SDMT of cognitive tests [64]. T2-lesion load (T2-LL) was recognized as an important predictor of memory function, cognitive efficiency and overall cognition [79]. Temporal, frontal and cerebellar hemispherical lesions had correlations with SDMT test performance, and a small cluster in left parietal with SDT. Inability of keeping recently learned information in memory was found to be correlated with lesions in superior parietal and left frontopolar and with adjacent regions of amygdalae and hippocampus [43]. White matter lesion volume (WMLV) was more strongly correlated with the cognitive tests (Paced Auditory Serial Addition Test (PASAT) and SDMT) compared to CL volume [32].

Decrease of thalamic volume was seen with a decrease in cognitive performance [52]. Normalized thalamic volume and anterior thalamic radiation integrity were among the predictors of cognitive decline [77, 79]. Both verbal and written parts of the SDMT test indicated moderate to strong correlations with the volume of thalamus nuclear groups [51].

In a study, CC density was another independent predictor of brief visuospatial memory test (BMVT) [49].

In RRMS, verbal memory performances correlated with atrophy of WM and left hippocampus [78]. Worse SDMT scores correlated with smaller normalized volume of the hippocampus and amygdala of each hemisphere and reduced R2t of the right hippocampus and amygdala, while worse performance on the 2s PASAT correlated with reduced R2t of the left amygdala [80]. The aforementioned alterations are indicative of cognitive impairment and therefore warrant the attention of medical professionals to evaluate potential cognitive decline in patients. However, it is important to note that these changes cannot be deemed as absolute predictors of cognitive impairment.

Lower cerebellar volumes, prominently posterior superior lobe (VI + Crus I) correlated with scores of SDMT and PASAT [72]. But these changes occurred in parallel with cognitive impairment and cannot predict it. An overview of included studies is shown in Table 1. We tried to analyze Table 1 with an aim to classify the results according to the region assessed, which ranged from cortical and gray matter, spinal cord, corpus callosum, cerebellum, thalamus, limbic system, lesion atrophy, brain volumetry, to lesions and white matter.

**Table 1. An overview of the literature regarding the studies with correlations of MRI markers with disability progression, progression of the disease and cognitive decline in studies participants.**

| Study | Demographic Features | | | Imaging Methodology | | Correlations with MRI markers |
|---|---|---|---|---|---|---|
| | Participants | Mean Age ± SD (years) or range | Mean Disease Duration ± SD (years) or range | F/S (T) | Parameter Studied | |
| Papadopoulou et al. 2013—C/S** [32] | 65 CIS and RRMS | 49.1 ± 1.85 | 17.4 ± 9.4 | 1.5 T | Cortical Lesion volume | **Disability:** Correlations between CL volume and EDSS. (r = 0.206; p = 0.051) **Cognition:** Assessment tools: SDMT, PASAT-3.CL volume was not predictor of neuropsychological outcomes. Subgroup analysis in RRMS: CLV only correlated with SDMT (r = -0.301, p = 0.019). In multiple regression analysis: CL load had no associations with any cognitive outcomes. |
| | 26 SPMS and PPMS | | | | | |
| Matsushita et al. 2018—C/S [33] | 13 CL group (9 RRMS + 1 PPMS + 3 SPMS) | 43.1 ± 10.1 | 13.8 ± 8.2 | 3 T | Cortical Lesion | **Disability:** Higher EDSS score in the CL group (2.8 ± 1.8) compared to non-CL group (0.5 ± 0.8); p = 0.009. **Progression:** In the non-CL group, all six patients had RRMS. In the CL group, 9 had RRMS, 1 had PPMS and 3 had SPMS. **Cognition:** No significant differences in the MMSE1 score, RCPM2 score, RCPM time, RBMT3 SPS4, RBMT SS5, TMT6-A, category WF7 task performance, visuospatial construction copying performance, letter WF task performance, visuospatial construction drawing performance and 1-s PASAT8, and 2-s PASAT between the CL and non-CL groups. The z-score analysis of the TMT and PASAT values were indicative of significant performance deterioration in CL group. |
| | 6 non-CL Group (6 RRMS) | 46.7 ± 9.4 | 16.7 ± 12.2 | | | |
| Haider et al. 2021—C [34] | 21 CIS | 60.5 ± 7.1 | 30.7 ± 1 | 3 T | Cortical Lesion | **Disability:** Highest EDSS scores were detected in 3 RRMS patients with cortical lesions. Cortical lesion counts (β: 0.37, 95% CI: 0.23 to 0.508), cervical spinal cord volume (β: −0.27, 95% CI: −0.421 to−0.109), grey matter volume (β:− 0.26, 95% CI:− 0.444 to− 0.074) collectively explained 60% (R2) of the variance of the EDSS In the model including only cortical lesions, 43% (R2) of the EDSS could be explained. Atrophy of GM was more predictive for EDSS. **Progression:** 30-year follow-up: the most important differences in MRI markers between SPMS and RRMS patients were the number of cortical lesions (cortical lesions had 88% specificity and 100% sensitivity), and GMV which was lower in SPMS. No cortical lesions detected in CIS, in 3 of 27 RRMS patients and in all SPMS patients. |
| | 27 RRMS | 60.6 ± 6.4 | 30.9 ± 1 | | | |
| | 15 SPMS | 61.9 ± 6.7 | 30.8 ± 0.9 | | | |
| Treaba et al. 2019—C [35] | 20 RRMS | 41.3 ± 10.5 | 6.0 ± 6.2 | 7 T | Cortical Lesion | **Disability:** Total CLV was identified as an independent predicting factor for baseline EDSS (β = 1.5, P = .001) and EDSS changes (β = 0.5, P = 0.003) at follow-up (near 1.5 years). **Progression:** Cortical lesion accumulation was bigger in SPMS than RRMS (3.6 lesions/ year ± 4.2 vs 1.1 lesions/year ± 0.9, respectively; P = 0.03) and preferentially localized in sulci. |
| | 11 SPMS | 39.9 ± 8.5 | 19.9 ± 9.0 | | | |
| | 10 HC | 39.9 ± 0.5 | | | | |

*(Continued)*

**Table 1.** (Continued)

| Study | Demographic Features | | | Imaging Methodology | | Correlations with MRI markers |
|---|---|---|---|---|---|---|
| | Participants | Mean Age ± SD (years) or range | Mean Disease Duration ± SD (years) or range | F/S (T) | Parameter Studied | |
| Kalinin et al. 2020—C/S [36] | ICL—Group (15 RRMS, 1 PPMS) | 27 (25–36.5) * | 15 (9–34.5) months * | 3 T | Intracortical lesions | **Disability:** Patients with intracortical lesions had higher EDSS scores (P = 0.02). |
| | ICL+ Group (39 RRMS, 12 SPMS, 4 PPMS) | 40 (32–53) * | 77 (26–171) months * | | | |
| Calabrese et al. 2010—C/S [37] | 76 RRMS | 34.2 | 4.8 | 1.5 T | Cortical Lesion volume | **Disability:** Baseline CLV correlated with baseline EDSS (r = 0.36, p = 0.001) and EDSS changes (r = 0.51, p = 0.001) over time. In both patient groups, baseline CLV was identified as an independent predicting factor for EDSS worsening and GM volume change at follow-up. **Progression:** Baseline CLs were detected more among SPMS (74.2%) than RRMS (64.4%). During follow-up, 0.8 new CL/patient/year in RRMS and 1.0 new CL/patient/year in SPMS were seen (all non-significant). Increase of CLV and number was significantly higher in the 52 clinically worsened patients compared to those who remained stable. |
| | 31 SPMS | 41.1 | 11.6 | | | |
| Scalfari et al. 2018—C [62] | 160 RRMS remained RRMS | 31.5 ± 10.6 | 7.8 ± 1.3 | 1.5 T | Cortical lesions number and thickness | **Progression:** Higher baseline cortical lesions significantly correlated with the higher risk of SPMS conversion (hazard ratio of 2.16 for 2 CLs, 4.79 for 5 CLs, and 12.3 for 7 CLs). |
| | 59 RRMS converted to SPMS | 34.2 ± 7.6 | 8.2 ± 1.0 | | | |
| Louapre et al. 2018—C/S [38] | 10 Early RRMS | 38.3 ± 10.5 | 2.8 ± 1.0 | 3 T—7 T | Cortical lesions volume | **Disability:** EDSS correlated positively with CLV (Adj $R^2$ = 30%). **Progression:** CLV was higher in SPMS compared to other subgroups of MS. **Cognition:** CLV was the most powerful, independent predictor, explaining 40% of the variance of SDMT. |
| | 18 RRMS | 44.3 ± 7.6 | 11.4 ± 4.2 | | | |
| | 13 SPMS | 45.5 ± 8.1 | 18.5 ± 8.4 | | | |
| | 17 HC | 39.3 ± 8.8 | NA | | | |
| Pinter et al. 2015—C/S [79] | 17 CIS | 33.1 ± 9.1 | 0.4 ± 0.8 | 3 T | Cortical volume | **Cognition:** In univariate analysis there was a positive effect of NCV (βj = 0.39; p<0.05) on overall cognition. Cognitive efficiency and overall cognitive function were strongly predicted by cortical volume. |
| | 47 RRMS | 35.8 ± 10.5 | 9.0 ± 8.7 | | | |
| | 5 SPMS | 41.8 ± 9.3 | 11.3 ± 6.0 | | | |
| Eijlers et al. 2018—C [77] | 168 Cognitively Stable (144 RRMS, 15 SPMS, 9 PPMS) | 46.7 ± 11.0 | 14.2 ± 8.3 | 3 T | Cortical gray matter volume | **Cognition:** Cortical GMV was defined as the only significant MRI predictor of cognitive decline in a whole-brain model (Nagelkerke R2 = 0.22, P<0.001). WM integrity damage was predictor of early RRMS cognitive decline, while in late RRMS and progressive MS, it was predicted by cortical atrophy. |
| | 66 Cognitively Declining (38 RRMS, 18 SPMS, 10 PPMS) | 49.7 ± 10.8 | 15.9± 8.4 | | | |
| | 60 HC | 46.4 ± 9.9 | NA | | | |
| Burgetova et al. 2017—C/S [54] | 80 RRMS | 46.9 ± 7.0 | 12.4 ± 10.7 | 1.5 T | Iron in deep gray matter | **Disability:** EDSS score was positively associated with iron accumulation in the putamen of RRMS and PPMS and caudate of RRMS patients. |
| | 40 EDSS matched RRMS | 48.6 ± 7.0 | 13.2 ± 11.0 | | | |
| | 24 PPMS | 47.4 ± 6.8 | 7.7 ± 3.3 | | | |
| | 20 HC | 48.0 ± 7.3 | NA | | | |

*(Continued)*

**Table 1.** (Continued)

| Study | Demographic Features | | | Imaging Methodology | | Correlations with MRI markers |
|---|---|---|---|---|---|---|
| | Participants | Mean Age ± SD (years) or range | Mean Disease Duration ± SD (years) or range | F/S (T) | Parameter Studied | |
| Rocca et al. 2021—C/S [39] | 34 CIS | 33.8 (19–50) | 0.5 (0.08–3) | 3 T | Gray matter volume | **Disability**: Lower NGMV (OR = 0.98, 95% CI = 0.96–0.99, p = 0.008) and lower GMV in the cerebellar network (OR = 0.40, 95% CI = 0.19–0.85, p = 0.01) are independent predictors of disability worsening (AUC = 0.83). |
| | 226 RRMS | 44.0 (18–70) | 12.7 (0.1–37) | | | |
| | 95 SPMS | 54.3 (33–72) | 21.5 (3–46) | | | |
| | 43 PPMS | 55.4 (27–77) | 15.8 (2–45) | | | |
| | 170 HC | 40.0 (19–75) | NA | | | |
| Tsagkas et al. 2021—C [40] | 140 RRMS | 43.8 ± 10.2 | 14.0 ± 8.7 | 1.5 T | Atrophy of gray matter | **Disability**: Higher baseline GM and GM AVCR in SPMS were associated with higher T25FWT deterioration (mean yearly decrease of 1/T25fwt of $-4.2 \times 10^{-3} \pm 1.7 \times 10^{-4}$; p = 0.087. Lower baseline GMV in RRMS was associated with higher D9HPT deterioration (mean yearly increase of log [D9HPT] of $0.012 \pm 2.0 \times 10^{-3}$, p = $7.5 \times 10^{-7}$) |
| | 43 SPMS | 55.0 ± 8.8 | 21.3 ± 9.2 | | | |
| Eshaghi et al. 2018—C [42] | 253 CIS | 33.0 ± 8.0 | 0.4 ± 1.4 | Various | Cortical gray matter and deep gray matter volume | **Disability:** Time-to-EDSS progression was predicted by DGM (hazard ratio = 0.73, 95% CI: 0.65, 0.82; p<0.001): for every SD decrease in baseline DGMV, there was a 27% higher risk of presenting a shorter EDSS progression time during follow-up. **Progression:** Lowest volumes of DGM and cortical GM at baseline were detected in SPMS. DGM showed the fastest annual rate of atrophy out of all imaging markers, prominently in SPMS (-1.45%) and RRMS (-1.34%) compared to CIS (-0.88%) and HCs (-0.94%) [p<0.01]. Temporal (- 1.21%) and parietal (-1.24%) GM atrophy was fastest in SPMS (All p values <0.05). |
| | 708 RRMS | 38.2 ± 9.8 | 6.7 ± 7.3 | | | |
| | 128 SPMS | 48.2 ± 9.8 | 15.6 ± 9.9 | | | |
| | 125 PPMS | 48.5 ± 10.1 | 6.8 ±5.9 | | | |
| | 203 HC | 38.7 ± 10.5 | NA | | | |
| Moccia et al. 2017—C [41] | 119 RRMS remained RRMS | 32.7 ± 7.4 | 4.2 ± 2.8 | Various | Gray: White matter ratio | **Disability:** EDSS worsening was associated with GM/ NAWM ratio (coefficient, 2.918; 95%CI, 4.739–1.097). Patients diagnosed with a higher GM/ NAWM ratio had a 90% lower rate of reaching EDSS 4.0 (hazard ratio, 0.111; 95% CI, 0.020–0.609) compared to patients with lower GM/ NAWM ratio. **Progression:** Higher baseline GM/NAWM ratio associated with lower rate of converting to SPMS (hazard ratio, 0.017; 95% CI, 0.001–0.203). |
| | 30 RRMS converted to SPMS | 34.2 ± 6.3 | 4.6 ± 2.7 | | | |
| Kearney et al. 2016—C/S [81] | 25 CIS | 36.5 ± 9.0 | 0.4 ± 0.4 | 3 T | Spinal Cord Focal lesions (GM & WM) | **Disability:** Lesion number per patient in both the lateral column and expanding to gray matter had independent associations with disability (p < 0.001). **Progression:** Percentage of patients with focal lesions involving at least two WM columns and expanding to gray matter was higher in SPMS compared to RRMS (p = 0.03) and PPMS (p = 0.015). Diffuse abnormalities were more common in both PPMS and SPMS, compared with RRMS (OR 6.1 (p = 0.002) and OR = 5.7 (p = 0.003), respectively). |
| | 35 RRMS | 38.7 ± 9.7 | 6.5 ± 5.2 | | | |
| | 30 PPMS | 50.6 ± 9.9 | 10.4 ± 7.5 | | | |
| | 30 SPMS | 51.1 ± 9.2 | 19.9 ± 11.5 | | | |

(*Continued*)

**Table 1.** (Continued)

| Study | Demographic Features | | | Imaging Methodology | | Correlations with MRI markers |
|---|---|---|---|---|---|---|
| | Participants | Mean Age ± SD (years) or range | Mean Disease Duration ± SD (years) or range | F/S (T) | Parameter Studied | |
| Kantarci et al. 2016—C [82] | 324 RIS remained RRMS | 38.6 (14–74) * | NA | Various | Spinal cord lesions | **Progression**: PPMS cases had more SC lesions (100%) than CIS/MS cases (64%) and asymptomatic cases (23%) within the follow-up period (P = 0.005). |
| | 113 RIS converted to CIS/M | 32 (11–70) * | NA | | | |
| | 15 RIS converted to PPMS | 43.3 (20–66) * | NA | | | |
| Nakamura et al. 2020—C/S [55] | 111 RRMS | 43.9 ± 12.7 | 13.8 ± 10.1 | 1.5 T- 3 T | Spinal cord area | **Disability:** CS-SCA at all four levels was negatively correlated with EDSS scores (P < 0.0001 at C2/C3 and C3/C4, P = 0.0002 at T8/T9, and P = 0.002 at T9/T10). And FSS for pyramidal symptoms (P = 0.0002 at C2/C3, P < 0.0001 at C3/C4 and T8/T9, and P = 0.0005 at T9/T10). Cervical CS-SCA was more strongly correlated with EDSS compared to thoracic CS-SCA. **Progression:** Cervical CS-SCA is smaller in PMS than in RRMS (mean 57.0 vs 61.0 mm2, P = 0.02 at C2/C3, and mean 58.8 vs 63.4 mm2, P = 0.007 at C3/C4). (As predictive factor). |
| | 29 Progressive MS | 49.3 ± 12.2 | 16.6 ±8.7 | | | |
| Bernitsas et al. 2015—C/S [56] | 93 RRMS | 39.3 ± 7.9 | 9.3 ± 3.3 | 3 T | Cross sectional area of cervical cord at C2 vertebral level | **Disability:** There was a correlation between CSA-C2 and EDSS (r = -0.75, P<0.0001). CSA-C2 was a predictor of disability independent of disease duration, and phenotype. Sub-group analysis showed a modest inverse relationship between the CSA-C2 and EDSS in the RRMS (-0.38, p = 0.0004) and progressive groups (0.4, p = 0.0021) **Progression:** CSA-C2 volume loss was more prominent in PMS compared to RRMS (68.6 ± 7.4mm2 vs. 87.3 ±8.4 mm2, p<0.0001), consistent with the neurological disability of both. |
| | 57 PMS | 44.5 ± 8.3 | 14.4 ± 4.5 | | | |
| Bonacchi, et al. 2020—C/S [57] | 58 RRMS | 42.0 ± 10.0 | 8 (2–16) * | 3 T | Cervical spinal cord | **Disability:** CSC GM-CSA is predictor of EDSS in PMS (R2 = 0.44) and RRMS (R2 = 0.51). RRMS: EDSS is associated with CSC global and regional normalized T2 lesion volume. (P values ranged from 0.02 to 0.002, except for the dorsal column, with P> 0.05) PMS: EDSS is associated with CSC GM (P = 0.003) and WM atrophy (P = 0.02). **Progression:** CSC-GM atrophy (CSA <11.1 mm2) was defined as a precise predicting factor for progressive phenotype. |
| | 62 PMS | 50.0 ± 10.0 | 18 (9–24) * | | | |
| | 30 HC | 43.0 ± 14.0 | NA | | | |
| Rocca et al. 2013—C/S [71] | 15 CIS | 29.1 ± 7.7 | 0.04 (0.01-0.08) | 1.5 T | Cervical spinal cord atrophy and lesion | **Disability:** SC atrophy at C1/C2 had correlations with the pyramidal FS score in SPMS (r = −0.91, p<0.001 uncorrected) and PPMS (r = −0.89, p<0.001 uncorrected) and with EDSS in PPMS (r = −0.68, p<0.001 uncorrected). **Progression:** PPMS had significant cord atrophy. |
| | 15 RRMS | 39.2 ± 11.4 | 4.3 (1–10) | | | |
| | 15 PPMS | 43.9 ± 7.0 | 22.6 (15–39) | | | |
| | 13 SPMS | 47.6 ± 8.7 | 16.8 (5–34) | | | |
| | 19 BMS | 43.9 ± 11.1 | 6.7 (1–14) | | | |
| | 31 HC | 39.9 ± 12.6 | NA | | | |

*(Continued)*

**Table 1.** (Continued)

| Study | Demographic Features | | | Imaging Methodology | | Correlations with MRI markers |
|---|---|---|---|---|---|---|
| | Participants | Mean Age ± SD (years) or range | Mean Disease Duration ± SD (years) or range | F/S (T) | Parameter Studied | |
| Tsagkas et al. 2018—C [58] | 180 RRMS | 41.5 ± 10.1 | 11.4 ± 8.4 | 1.5 T | Spinal cord volume loss | **Disability:** SCV loss was a strong predictor of EDSS score worsening (p <0.05). In RRMS and SPMS, mean annual rate of spinal cord volume loss was identified as the strongest prediction factor of mean annual EDSS alteration. **Progression:** The mentioned predictive role for spinal cord volume loss in the previous column is stronger in SPMS than RRMS. |
| | 51 SPMS | 55.4 ± 7.6 | 191. ± 9.7 | | | |
| Tsagkas et al. 2019—C [60] | 12 PPMS | 46.7 ± 6.6 | 8.6 ± 7.1 | 1.5 T | Upper cervical Spinal cord volume | **Disability:** Both PPMS and RRMS were associated with average EDSS over years, but only PPMS was associated with EDSS increase over time. **Progression:** Spinal cord volume loss was higher in PPMS compared to SPMS (p = 0.066) and RRMS (p < 0.01) over time. |
| | 24 RRMS | 48.2 ± 9.6 | 8.9 ± 5.9 | | | |
| | 24 SPMS | 50.6 ± 7.4 | 12.3 ± 6.6 | | | |
| Lukas et al. 2015—C [59] | 256 RRMS | 41.1 (35–48.1) * | 8 (4–15) * | 1.5 T | Spinal cord atrophy and lesion | **Disability:** Baseline aUCCA and the number of SC segments affected by T2-lesions were the most associated MRI markers indicative of EDSS worsening. **Progression:** All MS types had reduction of UCCA over 2 years. The annualized 24-month reduction rate of UCCA was higher in SPMS than in RRMS (p = 0.019) but did not discriminate PPMS from either RRMS or SPMS. Baseline UCCA was lower in SPMS than PPMS and RRMS. SC lesion numbers and SC affected segment numbers were higher in SPMS than PPMS and RRMS (P≤ 0.01). (SPMS was worse than other types). |
| | 73 SPMS | 54.2 (48–59.4) * | 18 (12.5–26) * | | | |
| | 23 PPMS | 49 (44.8–56) * | 6 (4–13) * | | | |
| Bischof et al. 2022—C [70] | 147 RRMS remained stable | 41 (18) * | 5 (10) * | 3 T | Cervical cord area at C1 level (C1A) | **Disability:** No association was found between C1A atrophy rates and baseline EDSS during the period of pre conversion (0.16%/year,0.50 to 0.17, p = 0.334). **Progression:** Patients who converted to SPMS indicated faster cord atrophy (-2.19% per year) compared to their RRMS matches (-0.88% per year) at least 4 years before conversion. The rate of cord atrophy was decreased after conversion (-1.63%/year, p = 0.010). Each 1% faster spinal cord atrophy rate was associated with 53% and 69% shorter time to SPMS conversion and silent progression, respectively. |
| | 159 RRMS with silent progression | 41 (13) * | 6 (10) * | | | |
| | 47 SPMS | 47 (14) * | 17 (14) * | | | |
| | 80 HC | 41 (18) * | NA | | | |
| Haider et al. 2021—C [34] | 21 CIS | 60.5 ± 7.1 | 30.7 ± 1 | 3 T | Cervical spinal cord volume | **Disability**: Cervical cord volume was associated with EDSS (β: –0.27, 95% CI: –0.421 to –0.109). |
| | 27 RRMS | 60.6 ± 6.4 | 30.9 ± 1 | | | |
| | 15 SPMS | 61.9 ± 6.7 | 30.8 ± 0.9 | | | |

*(Continued)*

**Table 1.** (Continued)

| Study | Demographic Features | | | Imaging Methodology | | Correlations with MRI markers |
|---|---|---|---|---|---|---|
| | Participants | Mean Age ± SD (years) or range | Mean Disease Duration ± SD (years) or range | F/S (T) | Parameter Studied | |
| Tsagkas et al. 2021—C [40] | 140 RRMS | 43.8 ± 10.2 | 14.0 ± 8.7 | 1.5 T | Atrophy of spinal cord | **Disability**: SPMS: Higher CSC AVCR was associated with EDSS (Mean yearly increase of log [EDSS] of $0.024 ± 5.2 × 10{-3}$, p = $6.7 ×10{-5}$) and future T25FWT (Mean yearly decrease of 1/T25fwt of -$4.2 × 10{-3} ± 1.7 × 10{-4}$; p = 0.087) worsening over time. |
| | 43 SPMS | 55.0 ± 8.8 | 21.3 ± 9.2 | | | |
| Yaldizli et al. 2010—C [67] | 169 MS (145 RRMS, 24 SPMS) | 42.0 ± 11.3 | 10.9 ± 8.8 | 1.5 T | Corpus callosum index | **Disability:** Corpus callosum index as a correlate of brain atrophy, was associated with disability progression. Although, it was not a long-term independent predictor. CCI at diagnosis was 0.345 ± 0.04 and correlated with EDSS at diagnosis. After a follow-up of 7.1 ±6.4 years, last EDSS correlated with CCI at diagnosis and last MRI (p = 0.002; r = 0.283 and p<0.001; r = 0.301 respectively) **Progression:** 24 patients had secondary progression with lower corpus callosum index values at diagnosis compared to those remained unchanged (0.308 ± 0.08 in SPMS vs.0.353 ± 0.06 in RRMS, p = 0.003). Corpus callosum index decrease in SPMS was two times more than RRMS (p = 0.04). |
| Petracca et al. 2020—C/S [49] | 13 RRMS | 46.8 ± 11.2 | 14.9 ± 8.5 | 3 T | Streamline density and focal lesions in corpus callosum sub regions | **Disability:** Corpus callosum density was identified as an independent predictor of 9-HPT (β = -0.327, p = 0.018), T25FWT (β = -0.357, p = 0.021) and EDSS (β = -0.328, p = 0.018). Corpus callosum damage was a predictor of ambulation performance, global disability and manual dexterity. **Progression:** Streamline density decrease was distinguished in SPMS in all Corpus callosum sub-regions, in PPMS in posterior and mid-posterior corpus callosum and in RRMS, only in posterior corpus callosum. **Cognition:** Corpus callosum density was independent predictor of BVMT (β = 0.344, p = 0.023). |
| | 20 SPMS | 55.3 ± 8.4 | 23.0 ± 13.6 | | | |
| | 22 PPMS | 52.3 ± 9.7 | 8.9 ± 5.2 | | | |
| | 24 HC | 46.4 ± 10.5 | NA | | | |
| Uher et al. 2019—C [50] | 386 CIS | 33.8 ± 9.0 | 3.0 ± 5.2 | 1.5 T | Corpus callosum volume | **Disability**: All patients with annualized percent corpus callosum volume change (cut-off < -0.39%) had higher EDSSAAC (p ≤ 0.001–0.035), except for CIS (p = 0.09–0.29). |
| | 964 RRMS | 35.3 ± 8.5 | 8.2 ± 6.5 | | | |
| | 63 SPMS | 40.9 ± 9.1 | 12.6 ± 6.8 | | | |
| | 58 HC | 37.5 ± 9.1 | NA | | | |
| D' Ambrosio et al. 2017—C/S [72] | 52 RRMS | 43.3 ± 11.2 | 8.2 (0–34) * | NA | Cerebellar subregions | **Disability:** Anterior cerebellar volume (lobules I—V) was identified as an independent predicting factor for EDSS and 9-HPT performance. **Progression:** Lower cerebellar volumes were observed in SPMS compared to BMS and RRMS patients (total and anterior cerebellar volume). **Cognition:** Cognitive performance (SDMT and PASAT scores) showed correlations with lower cerebellar volumes, prominently posterior lobe (lobules VI—X). |
| | 20 BMS | 42.6 ± 7.8 | 18.4 (15–26) * | | | |
| | 23 SPMS | 51.9 ± 9.1 | 19.6 (3–42) * | | | |
| | 32 HC | 39.6 ± 8.4 | NA | | | |

(*Continued*)

**Table 1.** (Continued)

| Study | Demographic Features | | | Imaging Methodology | | Correlations with MRI markers |
|---|---|---|---|---|---|---|
| | Participants | Mean Age ± SD (years) or range | Mean Disease Duration ± SD (years) or range | F/S (T) | Parameter Studied | |
| Parmar et al. 2022—C [73] | 125 RRMS | 44.7 ± 10.9 | 14.5 ± 10.9 | 1.5 T | Cerebellar volume | **Disability:** In RRMS, cerebellar volumes significantly predicted average EDSS, T25FWT and 9HPT. Atrophy of motor-related lobules (IV-VI + VIII) significantly predicted future worsening of non-dominant hand 9HPT. **Progression:** SPMS patients showed faster volume loss of posterior superior lobe compared to RRMS. **Cognition:** In RRMS, cerebellar volumes predicted SDMT. In SPMS, the atrophy rate of the posterior superior lobe (VI + Crus I) predicted future PASAT performance worsening. |
| | 38 SPMS | 55.1 ± 8.8 | 21.5 ± 9.7 | | | |
| Varoḡlu et al. 2010—C/S [74] | 14 RRMS | 29.0 ± 11.0 | 2.93 ± 2.95 | 1.5 T | Cerebellar volume | **Disability:** Cerebellum volume and EDSS showed significant correlations in both RRMS and SPMS. **Progression:** The mean cerebellum volume was decreased in all MS patients (RRMS and SPMS together) compared to controls (129. ± 4.79 cm3, p = 0.004) and (153.4 ± 6.55 cm3, respectively; p < 0.001). The mean cerebellum volume in RRMS was higher (137 ± 5.42 cm3) than SPMS (122 ± 4.34 cm3) (p< 0.001). |
| | 13 SPMS | 38.0 ± 11.0 | 8.61 ± 3.33 | | | |
| | 26 HC | 33.0 ± 8.0 | NA | | | |
| Calabrese et al. 2010—C/S [75] | 38 CIS | 37.0 ± 8.5 | 0.4 ± 0.4 | NA | Cerebellar cortical lesions and volume | **Disability:** The CCV (β = -0.601, p<0.001) and the cerebellar CL volume (β = 0.512, p<0.001) were the best predictors of cerebellar disability. CCV was identified as independent predictor of EDSS (b = -0.339, p = 0.011). **Progression:** Lowest CCV values were observed in PPMS. All MS subgroups had high reduction in CCV, except for CIS. Significantly higher number and volume of cerebellar CL were observed in SPMS and PPMS compared to RRMS and CIS (p<0.001 for both comparisons). |
| | 35 RRMS | 37.8 ± 7.5 | 6.8 ± 6.5 | | | |
| | 27 SPMS | 43.6 ± 9.2 | 10.5 ± 7.4 | | | |
| | 25 PPMS | 45.5 ± 6.2 | 8.4 ± 6.4 | | | |
| | 32 HC | 35.9 ± 7.5 | NA | | | |
| Petracca et al. 2022—C [83] | 838 RRMS | 37.7 ± 9.6 | 4.2 ± 5.5 | NA | Cerebellar lesions number | **Disability**: Shorter 9HPT deterioration time was associated with anterior cerebellar volume (p = 0.0444) and higher cerebellar T2 lesions volume (HR = 2.211, p = 0.0002). CDP showed associations with volume and number of cerebellar Gd+ lesions (p = 0.0389 and p = 0.0223, respectively). |
| Favaretto et al. 2016—C/S [76] | 10 CIS | 30.0 ± 7.2 | 0.9 ± 0.6 | 3 T | Cerebellar cortical lesions | **Disability:** Cerebellar CL number was highly correlated with EDSS in both Double Inversion Recovery (r = 0.69, p<0.0001) and Phase Sensitive Inversion Recovery (r = 0.72, p<0.0001). **Progression:** CL was observed in 26 patients by Double Inversion Recovery and in 31 by Phase Sensitive Inversion Recovery, and their number increased from CIS/eRRMS to SPMS (p = 0.001). |
| | 24 RRMS | 40.9 ± 7.2 | 9.6 ± 7.0 | | | |
| | 6 SPMS | 44.5 ± 9.8 | 20.2 ± 10.9 | | | |

*(Continued)*

**Table 1.** (Continued)

| Study | Demographic Features | | | Imaging Methodology | | Correlations with MRI markers |
|---|---|---|---|---|---|---|
| | Participants | Mean Age ± SD (years) or range | Mean Disease Duration ± SD (years) or range | F/S (T) | Parameter Studied | |
| Trufanov et al. 2021—C/S [51] | 40 RRMS | 31.7 ± 5.9 | 2.3 ±1.5 | 3 T | Thalamus nuclei volumes | **Disability:** EDSS had a negative correlation with the volumes of left pulvinar nuclei and the medial nucleus of right pulvinar. **Progression:** Significant differences were seen between RRMS and SPMS, in volumes of left side medial and lateral geniculate nuclei, lateral dorsal and anterior ventral nuclei. MR-morphometry of dominant deep thalamic nuclei were considered as a key predictor of MS progression. **Cognition:** Both sections (verbal and written) of the SMDT test showed moderate to strong correlations with the nuclei of thalamus, prominently with those on left side (r > 0.4). Written section of SDMT had the highest correlation with the left ventral anterior nucleus (r = 0.71). |
| | 28 SPMS | 33.3 ±5.7 | 5.5 ±4.4 | | | |
| | 10 HC | NA | NA | | | |
| Azevedo et al. 2018—C [52] | 520 MS (90 CIS, 392 RRMS, 38 SPMS) | 42.7 ± 9.8 | 9.2 ± 8.6 | 3 T | Thalamic volume | **Disability:** Thalamic volume reduction inversely associated with increase of EDSS (r = -0.29, CI: -0.21, -0.37) p<0.01), 9-HPT (r = -0.37, CI: -0.30, -0.45, p<0.01) and timed 25-FWT (r = -0.25, CI: -0.16, -.32, p<0.01). **Cognition:** Thalamic volume reduction associated with decrease of MSFC (r = 0.32, CI: 0.24, 0.40, p<0.01) and PASAT (r = 0.15, CI: 0.06, 0.23, p<0.01). |
| | 81 HC | 41.1 ± 9.7 | NA | | | |
| Rocca et al. 2010—C/S [68] | 20 CIS | 28.2 ± 4.9 | NA | 1.5 T | Thalamic lesions volume | **Progression:** SPMS patients has significantly higher baseline mean T1 lesion volume and higher change of volume compared to RRMS (P < .001 and .002, respectively). |
| | 34 RRMS | 32.7 ± 8.4 | 7 (2–25) * | | | |
| | 19 SPMS | 40.5 ± 10.6 | 8 (3–23) * | | | |
| | 13 HC | 33.3 | NA | | | |
| Louapre et al. 2018—C/S [38] | 10 Early RRMS | 38.3 ± 10.5 | 2.8 ± 1.0 | 3 T- 7 T | Thalamic lesions volume | **Progression:** Higher thalamic lesion volume was observed in SPMS compared to RRMS (0.16 cm3 vs 0.01 cm3). The most decreased thalamic volume was found in SPMS subjects. |
| | 18 RRMS | 44.3 ± 7.6 | 11.4 ± 4.2 | | | |
| | 13 SPMS | 45.5 ± 8.1 | 18.5 ± 8.4 | | | |
| | 17 HC | 39.3 ± 8.8 | NA | | | |
| Magon et al. 2020—C [53] | 179 RRMS | 41.4 ± 10.2 | 11.3 ± 8.3 | 1.5 T | Volume loss in thalamic subnuclei | **Disability:** EDSS change was associated with anterior and ventral anterior (in MS and RRMS), and pulvinar (in MS) nucleus volume loss. Annual rates of thalamus and ventral lateral nucleus volume loss were predictive of disability worsening. **Progression:** Significant predictors of disease progression were baseline volume and annual rate of ventral lateral nucleus volume loss. Every 1% increase of the annual rate of volume loss was associated with a 20% higher risk of disease progression in the following year. |
| | 50 SPMS | 55.3 ± 7.7 | 18.7 ± 9.6 | | | |
| Burgetova et al. 2017—C/S [54] | 80 RRMS | 46.9 ± 7.0 | 12.4 ± 10.7 | 1.5 T | Thalamic iron content, lesion load, brain parenchymal fraction | **Disability:** EDSS was negatively associated with thalamic iron. **Progression:** RRMS patients had significantly lower regional susceptibility in thalamus compared to PPMS group (P = 0.007). |
| | 40 EDSS matched RRMS | 48.6 ± 7.0 | 13.2 ± 11.0 | | | |
| | 24 PPMS | 47.4 ± 6.8 | 7.7 ± 3.3 | | | |
| | 20 HC | 48.0 ± 7.3 | NA | | | |

*(Continued)*

**Table 1.** (Continued)

| Study | Demographic Features | | | Imaging Methodology | | Correlations with MRI markers |
|---|---|---|---|---|---|---|
| | Participants | Mean Age ± SD (years) or range | Mean Disease Duration ± SD (years) or range | F/S (T) | Parameter Studied | |
| Tavazzi et al. 2020—C/S [64] | 20 CIS | 42.6 ± 10.7 | 5.1 ± 5.7 | 3 T | Thalamic volume | **Progression**: RRMS patients had significantly higher percentage of thalamic volume change compared to PMS (- 6.6% vs -4.4%, p = 0.029). |
| | 85 RRMS | 44.6 ± 10.8 | 13.3 ± 8.9 | | | |
| | 42 PMS | 56.1 ± 6.3 | 22.5 ± 10.5 | | | |
| Eijlers et al. 2018—C [77] | 168 Cognitively Stable (144 RRMS, 15 SPMS, 9 PPMS) | 46.77 ± 11.02 | 14.2 ± 8.3 | 3 T | Thalamic radiations | **Cognition**: Anterior thalamic radiation integrity is as a significant predictor for cognitive decline (Nagelkerke R2 = 0.35, P<0.01). |
| | 66 Cognitively Declining (38 RRMS, 18 SPMS, 10 PPMS) | 49.77 ± 10.80 | 15.9 ± 8.4 | | | |
| | 60 HC | 46.45 ± 9.91 | NA | | | |
| Pinter et al. 2015—C/S [79] | 17 CIS | 33.1 ± 9.1 | 0.4 ± 0.8 | 3 T | Thalamic volume | **Cognition**: Normalized thalamic volume strongly predicted memory function in patients. |
| | 47 RRMS | 35.8 ± 10.5 | 9.0 ± 8.7 | | | |
| | 5 SPMS | 41.8 ± 9.3 | 11.3 ± 6.0 | | | |
| Uher et al. 2019—C [50] | 386 CIS | 33.8 ± 9.0 | 3.0 ± 5.2 | 1.5 T | Thalamic volume | **Disability**: Patients with thalamic volume loss (cut-off < -0.56) had higher EDSSAAC (p ≤ 0.001–0.035). |
| | 964 RRMS | 35.3 ± 8.5 | 8.2 ± 6.5 | | | |
| | 63 SPMS | 40.9 ± 9.1 | 12.6 ± 6.8 | | | |
| | 58 HC | 37.5 ± 9.1 | NA | | | |
| Tsagkas et al. 2021—C [40] | 140 RRMS | 43.8 ± 10.2 | 14.0 ± 8.7 | 1.5 T | Thalamic volume | **Disability**: Lower baseline thalamic volumes and higher future T25fwt deterioration were significantly associated in RRMS patients (mean yearly decrease of 1/T25fwt of -6.4 × 10−4±3.6 ×10−4, p = 0.095). |
| | 43 SPMS | 55.0 ± 8.8 | 21.3 ± 9.2 | | | |
| Sacco et al. 2015—C/S [78] | 26 Cognitively Preserved RRMS | 40.0 ± 5.8 | 12.0 ± 7.0 | 3 T | Hippocampal volume | **Progression:** Significant atrophy of both hippocampus, WM and GM was observed in RRMS compared to HC (p = 0.001). **Cognition:** Atrophy of left hippocampus and WM had correlations with verbal memory performances in RRMS. In the CI subgroup, verbal memory tests significantly correlated with atrophy of left hippocampus [LTS (r = 0.46; p = 0.04), CLTR (r = 0.55; p = 0.01)]. |
| | 20 Cognitively impaired RRMS | 39.1 ± 9.8 | 11.3 ± 6.1 | | | |
| | 25 HC | 36.3 ± 9.2 | NA | | | |
| Wen et al. 2017—C/S [80] | 32 RRMS | 54.2 ± 9.7 | NA | 3 T | Limbic system volume loss and tissue integrity | **Disability:** Reduced R2t of right amygdala correlated with worse EDSS scores (r = -0.29, p = 0.05). **Progression:** Both hippocampus and amygdala in SPMS had reduced R2t and NV compared to HC, except R2t in left amygdala. SPMS had reduced R2t of right amygdala and NV of both hippocampi compared to RRMS. PPMS patients had smaller NV in both hippocampi. **Cognition:** There was a moderate correlation of reduced R2t of the right hippocampus and amygdala with deteriorated SDMT and in left amygdala with worse performance on the 2s PASAT. Smaller NV of hippocampus and amygdala of each sides had moderate correlation with worse SDMT. |
| | 32 SPMS | 57.1 ± 9.5 | NA | | | |
| | 16 PPMS | 55.3 ± 9.3 | NA | | | |
| | 31 HC | 49.5 ± 15.9 | NA | | | |

(*Continued*)

**Table 1.** (Continued)

| Study | Demographic Features | | | Imaging Methodology | | Correlations with MRI markers |
|---|---|---|---|---|---|---|
| | Participants | Mean Age ± SD (years) or range | Mean Disease Duration ± SD (years) or range | F/S (T) | Parameter Studied | |
| Dwyer et al. 2018—C [61] | 18 CIS | 44.8 ± 11.0 | 3.9 ± 3.8 | 3 T | Atrophied lesion volume | **Disability:** Atrophied lesion volume had significant associations with disability progression (EDSS change) and follow-up 9HPT in both RRMS and PMS. **Progression:** Highest lesion atrophy was observed in PMS patients (P = .02). |
| | 126 RRMS | 43.8 ± 11.1 | 12.3 ± 8.4 | | | |
| | 48 PMS | 55.5 ± 7.9 | 22.8 ± 10.3 | | | |
| Tavazzi et al. 2020—C/S [64] | 20 CIS | 42.6 ± 10.7 | 5.1 ± 5.7 | 3 T | Atrophied lesion volume | **Progression:** Atrophied T2-LV of brain predicted worsening over time (p = .007). **Cognition:** In the progressive group, atrophied T2-LV was associated with follow-up SDMT (p = 0.003). |
| | 85 RRMS | 44.6 ± 10.8 | 13.3 ± 8.9 | | | |
| | 42 PMS | 56.1 ± 6.3 | 22.5 ± 10.5 | | | |
| Kizlaitienė et al. 2017—C/S [66] | 43 RRMS | 33.6 ± 9.2 | 7.5 ± 5.7 | 1.5 T | Brain atrophy (11 linear MRI measures and 7 indexes) | **Progression:** Significant discriminative MRI atrophy measurements higher in SPMS vs. RRMS are as follows: Frontal atrophy Index, EVANS index, Huckman Index, Bicaudatus Index, Width of third ventricle. For differentiating RRMS from SPMS Remission-Progression Index formula can be used: Remission-Progression Index = (RAVLT 1–5 SUM + DSST)/Huckman Index. **Cognition:** Brain atrophy had correlations with impairment of verbal memory and other neurocognitive symptoms. Correlation was found in bicaudatus ratio: with DSST and RAVLT1-5 SUM and width of the third ventricle. Significant discriminative Cognitive test results higher in RRMS vs. SPMS are as followed: RAVLT 1–5 SUM, DSST, DSB, FPT, ROCFT-copy, LFT-D, LFT-A, LFT-S, CATflT, IST, Story, WPA-1, WPA-2, TMTA and TMTB tests |
| | 45 SPMS | 47.8 ± 7.7 | 18.5 ± 7.6 | | | |
| Uher et al. 2019—C [50] | 386 CIS | 33.8 ± 9.0 | 3.0 ± 5.2 | 1.5 T | Whole brain volume, gray matter volume | **Disability:** Patients with BVL (cut-off < -0.34%), GMVL (cut-off < -0.49), had higher EDSSAAC (p ≤ 0.001–0.035), except for GMVL cut-offs in the RRMS cohort (p = 0.09–0.29). |
| | 964 RRMS | 35.3 ± 8.5 | 8.2 ± 6.5 | | | |
| | 63 SPMS | 40.9 ± 9.1 | 12.6 ± 6.8 | | | |
| | 58 HC | 37.5 ± 9.1 | NA | | | |
| Ajitomi et al. 2022—C/S [44] | 69 RRMS | 39.2 ± 8.3 | 8 (4.2–13.3) * | 1.5 T | Third & lateral ventricle width, whole-brain volume | **Disability:** EDSS had significant correlation with whole-brain volume (rho = -0.52, p <0.0001), Bicaudate ratio and width of third and lateral ventricle (stronger). |
| | 16 PMS | 46.9 ± 9.4 | 17.7 (9.3–25) * | | | |
| Lukas et al. 2010—C [65] | 25 MS (No progression) | 33 (28–38) * | 1.4 (0.5–3.6) * | 1 T | Annualized percentage ventricular volume change | **Disability:** There was a correlation between annual EDSS change over the 5.5-year follow-up and aPVVC. **Progression:** Patients having progression over 5.5 years had higher aPVVC within first 2 years compared with non-progressive patients (4.76%/y) compared with patients without progression (3.23%/year, p = 0.02). Each 1% increase in aPVVC every year, increases the odds of progression 1.17 times. |
| | 29 MS (Progression) | 39 (31–47) * | 1.3 (0.5–3.3) * | | | |

(*Continued*)

**Table 1.** (Continued)

| Study | Demographic Features | | | Imaging Methodology | | Correlations with MRI markers |
|---|---|---|---|---|---|---|
| | Participants | Mean Age ± SD (years) or range | Mean Disease Duration ± SD (years) or range | F/S (T) | Parameter Studied | |
| Rocca et al. 2013—C/S [71] | 15 CIS | 29.1 ± 7.7 | 0.04 (0.01-0.08) * | 1.5 T | Brain volume | **Progression**: Compared to RRMS and BMS, SPMS had lower NBV (p ranging from 0.03 to <0.001). They also had lower NBV than PPMS (p = 0.001). |
| | 15 RRMS | 39.2 ± 11.4 | 4.3 (1–10) * | | | |
| | 15 PPMS | 43.9 ± 7.0 | 22.6 (15–39) * | | | |
| | 13 SPMS | 47.6 ± 8.7 | 16.8 (5–34) * | | | |
| | 19 BMS | 43.9 ± 11.1 | 6.7 (1–14) * | | | |
| | 31 HC | 39.9 ± 12.6 | NA | | | |
| Popescu et al. 2013—C [45] | 18 CIS | 29 (24–34) * | 0 (0–0) * | 1 T- 1.5 T | Brain atrophy and Lesion volume | **Disability:** Lesion volume change and central atrophy were predictors of EDSS in all patients and also predictors of EDSS and MSSS in ROMS group. Lesion volume at 1 year was a predictor of EDSS in RRMS and ROMS group. Baseline brain volume predicted EDSS in the CIS group. Whole brain atrophy was a predictor of EDSS and MSSS in PPMS group. **Progression:** There was an association between central atrophy and clinical progression over 5.5 years, in early RRMS. |
| | 97 RRMS | 35 (29–40) * | 5 (2–8) * | | | |
| | 69 SPMS | 46 (40–52) * | 11 (8–17) * | | | |
| | 77 PPMS | 52 (47–58) * | 10 (6–13) * | | | |
| Moodie et al. 2012—C [46] | 84 MS (57 RRMS, 20 SPMS, 7 PPMS) | 42.6 ± 8.6 | NA | 1.5 T | Whole brain atrophy, T2 & T1 lesions volume | **Disability**: There was an association between lower baseline T2LV and EDSS worsening. Lower baseline MRI disease severity (for MRDSS and other individual MRI markers) predicted EDSS worsening. |
| Preziosa et al. 2022—C [47] | 39 RRMS remained RRMS | 36.1 ± 9.7 | 9.2 ± 5.4 | 3 T | Slowly Expanding Lesions | **Disability:** T2 hyperintense lesion volume, lower NBV, presence of ≥ 4SELs, a higher proportion of lesions defined as SELs were associated with EDSS progression at 9.1-year follow-up. Higher PBVC between year 2 and baseline was associated with EDSS score worsening. Higher proportion of SELs among baseline lesions was independent predictor of EDSS worsening (C-index = 0.892). **Progression:** T2 hyperintense lesion number and volume and lower NBV had associations with SPMS conversion over 9.1 years. |
| | 13 RRMS converted to SPMS | 39.0 ± 9.5 | 11.7 ± 9.2 | | | |
| Pinter et al. 2015—C/S [79] | 17 CIS | 33.1 ± 9.1 | 0.4 ± 0.8 | 3 T | T2 lesion load | **Cognition**: T2-LL was a negative predicting factor for memory function, cognitive efficiency (βj = -0.38; p <0.001) and overall cognition (βj = -0.32; p < 0.001). |
| | 47 RRMS | 35.8 ± 10.5 | 9.0 ± 8.7 | | | |
| | 5 SPMS | 41.8 ± 9.3 | 11.3 ± 6.0 | | | |
| Filli et al. 2012—C/S [84] | 106 RRMS Center 1 | 43.1 ± 10.7 | 10 (12.3) * | 1.5 T | White matter lesion | **Progression:** SPMS was associated with higher regional probability of T1 (not T2) hypointense lesions in the callosal body, the corticospinal tract, and other tracts close to lateral ventricles compared to RRMS (p ≤ 0.03). |
| | 103 RRMS Center 2 | 43.6 ± 9.3 | 8 (9) * | | | |
| | 31 SPMS Center 1 | 53.4 ± 8.2 | 19 (14) * | | | |
| | 31 SPMS Center 2 | 52.1 ± 7.4 | 19 (13) * | | | |
| Kincses et al. 2011—C [43] | 26 CIS | 36.0 ± 10.4 | 0.23 ± 5.5 | 3 T | Lesion Probability Mapping | **Disability**: EDSS had correlations with lesion probability in the periventricular WM at left frontal horn and around posterior horns. |
| | 89 RRMS | 35.2 ± 9.4 | 6.68 ± 7.6 | | | |
| | 6 SPMS | 43.0 ± 8.7 | 10.3 ± 7.4 | | | |

(*Continued*)

**Table 1.** (Continued)

| Study | Demographic Features | | | Imaging Methodology | | Correlations with MRI markers |
|---|---|---|---|---|---|---|
| | Participants | Mean Age ± SD (years) or range | Mean Disease Duration ± SD (years) or range | F/S (T) | Parameter Studied | |
| Mostert et al. 2010—C [48] | 96 RRMS | 34 (27–41) * | 2 (0–7) * | 1 T- 1.5 T | T2 Lesion load and number | **Disability:** T2 LL and T2 lesion number had correlations with MSSS in RRMS. **Progression:** Higher T2 lesions number and load and confluent lesions increased disability progression risk in RRMS. |
| | 35 SPMS | 41 (32–49) * | 10 (3–21) * | | | |
| | 55 PPMS | 48 (39–55) * | 4 (2–9) * | | | |
| Papadopoulo u et al. 2013—C/S [32] | 65 CIS and RRMS | 49.1 ± 1.85 | 17.4 ± 9.4 | 1.5 T | White matter lesion volume | **Disability:** WMLV correlated with EDSS (r = .290, p = 0.005). **Cognition:** WM lesion volume had stronger correlations with the cognitive tests (PASAT (r = −0.361, p = 0.001), SDMT (r = 0.585, p<0.001)) compared to CL volume. |
| | 26 SPMS and PPMS | | | | | |
| Calabrese et al. 2010—C/S [37] | 76 RRMS | 34.2 | 4.8 | 1.5 T | White matter lesion volume | **Disability**: Baseline T2 WMLV independently predicted EDSS progression in SPMS. |
| | 31 SPMS | 41.1 | 11.6 | | | |
| Louapre et al. 2018—C/S [38] | 10 Early RRMS | 38.3 ± 10.5 | 2.8 ± 1.0 | 3 T- 7 T | White matter lesion volume | **Progression**: WMLV was higher in SPMS than other MS subgroups (SPMS = 13.4 vs. RRMS = 2.5 vs. early RR = 1.7). |
| | 18 RRMS | 44.3 ± 7.6 | 11.4 ± 4.2 | | | |
| | 13 SPMS | 45.5 ± 8.1 | 18.5 ± 8.4 | | | |
| | 17 HC | 39.3 ± 8.8 | NA | | | |
| Enzinger et al. 2011—C/S [85] | 62 RRMS remained RRMS | 33.9 ± 8.7 | 5.7 ± 6.2 | 1.5 T | Lesion load | **Progression**: Converters to SPMS had unchanged T2 lesion load and doubled T1 lesion load and black hole ratio from baseline to follow-up compared to those who remained RRMS. |
| | 22 RRMS converted to SPMS | 40.0 ± 9.4 | 8.5 ± 6.9 | | | |
| Sacco et al. 2015—C/S [78] | 26 Cognitively Preserved RRMS | 40.0 ± 5.8 | 12.0 ± 7.0 | 3 T | White matter atrophy, lesion load | **Cognition**: Atrophy of WM had correlations with verbal memory performances in RRMS. T2-LL and volume had correlations with executive functions, processing speed, visuo-spatial memory and sustained attention performances. Verbal memory tests significantly correlated with atrophy of WM in cognitively impaired subgroup [LTS (r = 0.46; p = 0.05), CLTR (r = 0.55; p = 0.01), D-SRT (r = 0.45; p = 0.05)]. |
| | 20 Cognitively impaired RRMS | 39.1 ± 9.8 | 11.3 ± 6.1 | | | |
| | 25 HC | 36.3 ± 9.2 | NA | | | |
| Dal-Bianco et al. 2021—C [63] | 24 Iron rim lesion group | 36.6 (18–53.6) * | 4.7 (0.9–28) * | 7 T | Iron rim lesions | **Progression**: The interaction term Time × IRL status was significant in SPMS (P = 0.034) but not RRMS (P = 0.153), implicating dissimilar volume dynamics of IRLs compared to non-IRLs in SPMS. |
| | 9 non- iron rim lesion group | 31.0 (21–62.6) * | 8.3 (1.1–32) * | | | |

*(Continued)*

**Table 1.** (Continued)

| Study | Demographic Features | | | Imaging Methodology | | Correlations with MRI markers |
|---|---|---|---|---|---|---|
| | Participants | Mean Age ± SD (years) or range | Mean Disease Duration ± SD (years) or range | F/S (T) | Parameter Studied | |
| Tsagkas et al. 2021—C [40] | 140 RRMS | 43.8 ± 10.2 | 14.0 ± 8.7 | 1.5 T | Lesion load | **Disability**: Higher lesion-load AAVC was associated with higher EDSS progression and for non-dominant hand, with higher ND9HPT progression in SPMS. In RRMS, higher lesion load AAVC was associated with higher ND9HPT progression for non-dominant hand. In SPMS, higher WM AVCR was associated with higher T25fwt progression and for nondominant hand, with higher ND9HPT progression. |
| | 43 SPMS | 55.0 ± 8.8 | 21.3 ± 9.2 | | | |

\* These values are presented as Median (Interquartile range) or Median (Range).

\*\* Study designs are shown as C/S: cross-sectional and C: cohort studies.

F/S (T): Field strength in Tesla unit.

**Scales and tests used for assessment of physical and cognitive disability and progression in MS patients:** EDSS: Expanded Disability Status Scale; SDMT: Symbol Digit Modalities Test; PASAT: Paced Auditory Serial Addition Test; MMSE: Mini-Mental State Examination; RCPM: Raven's Colored Progressive Matrices; RBMT: Rivermead Behavioral Memory Test; SPS: Standardized profile score; SS: Screening scores; TMT: Trail Making Test; WF: Word fluency; T25FWT: Timed 25-foot walk test; 9HPT: Nine-hole peg test; BVMT: Brief Visuospatial Memory Test; EDSSAAC: Annualized absolute EDSS change; MSFC: Multiple Sclerosis Functional Composite; MSSS: Multiple Sclerosis severity score; MRDSS: Magnetic resonance disease severity scale; RAVLT: Rey Auditory Verbal Learning Test; DSST: Digit Symbol Substitution Test.

**Abbreviations:** SD: Standard deviation; MS: Multiple Sclerosis; MRI: Magnetic Resonance Imaging; CIS: Clinically isolated syndrome; BMS: Benign MS; ROMS: Relapse-onset MS; RRMS: Relapsing-remitting MS; SPMS: Secondary progressive MS; PPMS: Primary progressive MS; PMS: Progressive MS; HC: Healthy controls; CL: Cortical lesion; ICL: Intra-cortical lesion; GMV: Gray matter volume; WM: White matter; WMLV: White matter lesion volume; NGMV: Normalized gray matter volume; AVCR: Annual volume change rate; DGM: Deep gray matter; NAWM: Normal appearing white matter; NV: Normalized volume; LV: Lesion volume; PVVC: Percentage ventricular volume change; SEL: Slowly expanding lesion; NBV: Normalized brain volume; IRL: Iron rim lesion; CCV: Cerebellar cortical volume; CS-SCA: Cross-sectional spinal cord area; CSA: Cross-sectional area; SC: Spinal cord; aUCCA: Annualized percentage upper cervical cord cross-sectional area change. PBVC: percentage brain volume change.

The evaluation of risk of bias was carried out for all the studies that were included in the analysis. Out of the total 26 cohort and 27 cross-sectional studies, only one study was found to have a high risk of bias [82], while the remaining studies were categorized as moderate or low. The detailed results of the quality assessment of the included studies are presented in Tables 2 and 3.

## Discussion

In recent years, MRI has emerged as a valuable tool for both the diagnosis and monitoring of MS. Extensive research has been conducted to identify predictive imaging biomarkers for MS, evaluating white and gray matter metrics to forecast disease progression. Despite being in use for almost four decades, MRI techniques are still evolving, and novel and classic metrics are being explored to improve the diagnostic process, treatment guidance, and prognosis. The significant volume of high-quality research conducted in this field of MS has enabled us to enhance our capability to correlate MRI scan outcomes with clinical evolution and pathological studies, and derive much-needed prognosis biomarkers from these data.

In this comprehensive review, we provided an insight into the potential of MRI markers to predict disability progression, disease progression, and cognitive decline in MS. The presence of lesions and alterations in certain structures of the CNS, including white matter and gray

**Table 2. JBI risk of bias assessment for cohort studies.**

| Author, year | Q1 | Q2 | Q3 | Q4 | Q5 | Q6 | Q7 | Q8 | Q9 | Q10 | Q11 | % Yes | Risk |
|---|---|---|---|---|---|---|---|---|---|---|---|---|---|
| Haider et al., 2021 | ✕ | ✓ | ✓ | ✓ | ✓ | ✓ | ✓ | ✓ | ✓ | ✓ | ✓ | 91 | Low |
| Treaba et al., 2019 | ✓ | ✓ | ✓ | ? | ? | ✕ | ✓ | ✓ | ✓ | ✓ | ✓ | 72 | Low |
| Scalfari et al., 2018 | ✓ | ✓ | ✓ | ✓ | ✓ | ✕ | ✓ | ✓ | ✓ | ✓ | ✓ | 91 | Low |
| Eijlers et al., 2018 | ? | ✓ | ✓ | ✓ | ✓ | ✕ | ✓ | ✓ | ✕ | ? | ✓ | 63 | Moderate |
| Tsagkas et al., 2021 | ✓ | ✓ | ✓ | ? | ? | ✓ | ✓ | ✓ | ✓ | ✓ | ✓ | 81 | Low |
| Eshaghi et al., 2018 | ? | ✓ | ✓ | ✓ | ✓ | ? | ✓ | ✓ | ? | ? | ✓ | 63 | Moderate |
| Moccia et al., 2017 | ✓ | ✓ | ✓ | ✓ | ✓ | ✕ | ✓ | ✓ | ✓ | ✓ | ✓ | 91 | Low |
| Kantarci et al., 2016 | ✕ | ? | ✓ | ? | ? | ✓ | ✓ | ✓ | ? | ? | ✓ | 45 | High |
| Tsagkas et al., 2018 | ✓ | ✓ | ✓ | ✓ | ✓ | ? | ✓ | ✓ | ✓ | ? | ✓ | 81 | Low |
| Tsagkas et al., 2019 | ✓ | ✓ | ✓ | ✓ | ✓ | ✕ | ✓ | ✓ | ✓ | ✓ | ✓ | 91 | Low |
| Lukas et al., 2015 | ✕ | ✓ | ✓ | ✓ | ✓ | ✕ | ✓ | ✕ | ? | ✕ | ✓ | 54 | Moderate |
| Bischof et al., 2022 | ✓ | ✓ | ✓ | ✓ | ✓ | ✓ | ✓ | ✓ | ✓ | ✓ | ✓ | 100 | Low |
| Yaldizli et al., 2010 | ✓ | ✓ | ✓ | ? | ? | ✕ | ✓ | ✓ | ✓ | ✓ | ✓ | 72 | Low |
| Uher et al., 2019 | ✕ | ✓ | ✓ | ✓ | ✓ | ✓ | ✓ | ✓ | ✓ | ? | ✓ | 81 | Low |
| Parmar et al., 2022 | ✓ | ✓ | ✓ | ? | ? | ✕ | ✓ | ✓ | ✓ | ✓ | ✓ | 72 | Low |
| Petracca et al., 2022 | ✓ | ✓ | ✓ | ? | ? | ✕ | ✓ | ✓ | ✓ | ✓ | ✓ | 72 | Low |
| Azevedo et al., 2018 | ✓ | ✓ | ✓ | ✓ | ? | ✓ | ✓ | ✓ | ✓ | ✓ | ✓ | 91 | Low |
| Magon et al., 2020 | ✓ | ✓ | ✓ | ✓ | ? | ? | ✓ | ✓ | ✓ | ✓ | ✓ | 81 | Low |
| Dwyer et al., 2018 | ✓ | ✓ | ✓ | ✓ | ✓ | ✕ | ✓ | ? | ? | ✓ | ✓ | 72 | Low |
| Lukas et al., 2010 | ✓ | ✓ | ✓ | ? | ? | ✕ | ✓ | ✓ | ✓ | ✓ | ✓ | 72 | Low |
| Popescu et al., 2013 | ? | ✓ | ✓ | ? | ? | ✕ | ✓ | ✓ | ✓ | ✓ | ✓ | 63 | Moderate |
| Moodie et al., 2012 | ✓ | ✓ | ✓ | ✓ | ✓ | ✓ | ✓ | ✓ | ✓ | ✓ | ✓ | 100 | Low |
| Preziosa et al., 2022 | ? | ✓ | ✓ | ? | ? | ✕ | ✓ | ✓ | ✓ | ✓ | ✓ | 63 | Moderate |
| Kincses et al., 2011 | ✓ | ✓ | ✓ | ✓ | ✓ | ✓ | ✓ | ✓ | ✓ | ✓ | ✓ | 100 | Low |
| Mostert et al., 2010 | ✓ | ✓ | ✓ | ? | ? | ✓ | ✓ | ✓ | ✓ | ✓ | ✓ | 81 | Low |
| Dal-Bianco et al., 2021 | ✓ | ✓ | ✓ | ✓ | ✓ | ? | ✓ | ✓ | ✓ | ✓ | ✓ | 91 | Low |

**Abbreviations:** JBI: Joanna Briggs Institute, '✓' indicates yes, '✕' indicates no and '?' indicates unclear.

**Q1.** Were the two groups similar and recruited from the same population? **Q2.** Were the exposures measured similarly to assign people to both exposed and unexposed groups? **Q3.** Was the exposure measured in a valid and reliable way? **Q4.** Were the confounding factors identified? **Q5.** Were strategies to deal with confounding factors stated? **Q6.** Were the groups/participants free of the outcome at the start of the study (or at the moment of exposure)? **Q7.** Were the outcomes measured in a valid and reliable way? **Q8.** Was the follow up time reported and sufficient to be long enough for outcomes to occur? **Q9.** Was follow up complete, and if not, were the reasons to loss to follow up described and explored? **Q10.** Were strategies to address incomplete follow up utilized? **Q11.** Was appropriate statistical analysis used?

**Note:** The risk of bias was ranked as high when the study reached up to 49% of "yes" scores, moderate when the study reached from 50 to 69% of "yes" scores, and low when the study reached more than 70% of "yes" scores.

matter, corpus callosum, thalamus, and spinal cord, have been found to have a significant impact on disability progression in individuals with MS.

The progression and conversion of CIS or RRMS to progressive phenotypes of MS is a major concern among physicians and patients alike. Prognostic factors for MS progression have been identified, including various biomarkers and MRI parameters such as cortical lesion, gray matter volume change, whole brain atrophy, corpus callosum index, thalamic volume change, and certain spinal cord markers, including the presence of lesions in spinal cord or alterations in its cross-sectional area.

On the other hand, cognitive impairment is a significant and prevalent change that can occur during the course of MS. Unfortunately, it has not been deemed a sign or symptom of MS attacks or MS progression and has not been included in McDonald's criteria until recently

**Table 3. JBI risk of bias assessment for cross-sectional studies.**

| Author, year | Q1 | Q2 | Q3 | Q4 | Q5 | Q6 | Q7 | Q8 | % Yes | Risk |
|---|---|---|---|---|---|---|---|---|---|---|
| Papadopoulou et al., 2013 | ✗ | ✗ | ✓ | ✓ | ? | ? | ✓ | ✓ | 50 | Moderate |
| Louapre et al., 2018 | ? | ✓ | ✓ | ✓ | ✓ | ✓ | ✓ | ✓ | 87 | Low |
| Matsushita et al., 2018 | ✗ | ✓ | ✓ | ✓ | ? | ? | ✓ | ✓ | 62 | Moderate |
| Kalinin et al., 2020 | ✓ | ✓ | ✓ | ✓ | ✓ | ✓ | ✓ | ✓ | 100 | Low |
| Calabrese et al., 2010, (Ref 37) | ? | ? | ✓ | ✓ | ? | ? | ✓ | ✓ | 50 | Moderate |
| Pinter et al., 2015 | ✗ | ✓ | ✓ | ✓ | ✓ | ✓ | ✓ | ✓ | 87 | Low |
| Burgetova et al., 2017 | ✓ | ? | ✓ | ✓ | ✓ | ✓ | ✓ | ✓ | 87 | Low |
| Rocca et al., 2021 | ✓ | ✓ | ✓ | ✓ | ✓ | ✓ | ✓ | ✓ | 100 | Low |
| Kearney et al., 2016 | ✗ | ? | ✓ | ✓ | ? | ? | ✓ | ✓ | 50 | Moderate |
| Nakamura et al., 2020 | ✓ | ✓ | ✓ | ✓ | ? | ? | ✓ | ✓ | 75 | Low |
| Bernitsas et al., 2015 | ✗ | ✓ | ✓ | ✓ | ✗ | ? | ✓ | ✓ | 62 | Moderate |
| Bonacchi, et al., 2020 | ✓ | ✓ | ✓ | ✓ | ✓ | ✓ | ✓ | ✓ | 100 | Low |
| Rocca et al., 2013 | ✓ | ✓ | ✓ | ✓ | ✓ | ✓ | ✓ | ✓ | 100 | Low |
| Petracca et al., 2020 | ✗ | ? | ✓ | ✓ | ✗ | ? | ✓ | ✓ | 50 | Moderate |
| D' Ambrosio et al., 2017 | ✓ | ✓ | ✓ | ✓ | ✓ | ✓ | ✓ | ✓ | 100 | Low |
| Varoğlu et al., 2010 | ? | ✗ | ✓ | ✓ | ? | ? | ✓ | ✓ | 50 | Moderate |
| Calabrese et al., 2010, (Ref 75) | ✗ | ✗ | ✓ | ✓ | ✓ | ✓ | ✓ | ✓ | 75 | Low |
| Favaretto et al., 2016 | ? | ✗ | ✓ | ✓ | ? | ? | ✓ | ✓ | 50 | Moderate |
| Trufanov et al., 2021 | ? | ✓ | ✓ | ✓ | ? | ? | ✓ | ? | 50 | Moderate |
| Rocca et al., 2010 | ✓ | ✓ | ✓ | ✓ | ? | ✓ | ✓ | ✓ | 87 | Low |
| Tavazzi et al., 2020 | ✓ | ✓ | ✓ | ✓ | ✓ | ✓ | ✓ | ✓ | 100 | Low |
| Sacco et al., 2015 | ✓ | ✓ | ✓ | ✓ | ✓ | ✓ | ✓ | ✓ | 100 | Low |
| Wen et al., 2017 | ✗ | ? | ✓ | ✓ | ✓ | ✓ | ✓ | ✓ | 75 | Low |
| Kizlaitienė et al., 2017 | ✓ | ✓ | ✓ | ✓ | ✓ | ✓ | ✓ | ✓ | 100 | Low |
| Ajitomi et al., 2022 | ✓ | ✓ | ✓ | ✓ | ? | ? | ✓ | ✓ | 75 | Low |
| Filli et al., 2012 | ✗ | ✓ | ✓ | ✓ | ? | ? | ✓ | ✓ | 62 | Moderate |
| Enzinger et al., 2011 | ✗ | ✓ | ✓ | ✓ | ✓ | ✓ | ✓ | ✓ | 87 | Low |

**Abbreviations:** JBI: Joanna Briggs Institute, '✓' indicates yes, '✗' indicates no and '?' indicates unclear.

**Q1.** Were the criteria for inclusion in the sample clearly defined? **Q2.** Were the study subjects and the setting described in detail? **Q3.** Was the exposure measured in a valid and reliable way? **Q4.** Were objective, standard criteria used for measurement of the condition? **Q5.** Were the confounding factors identified? **Q6.** Were strategies to deal with confounding factors stated? **Q7.** Were the outcomes measured in a valid and reliable way? **Q8.** Was appropriate statistical analysis used?

**Note:** The risk of bias was ranked as high when the study reached up to 49% of "yes" scores, moderate when the study reached from 50 to 69% of "yes" scores, and low when the study reached more than 70% of "yes" scores.

[23]. Despite its inevitability, cognitive impairment is often overlooked and not addressed with proper treatment and prophylaxis. Therefore, it is crucial to recognize the importance of cognitive impairment as a potential sign of disease progression and should be given the same level of attention as physical disability.

Multiple MRI parameters have been investigated as probable biomarkers for MS. Among these parameters, white matter lesions (WMLs) are a characteristic feature of MS, which are usually detected by contrast-enhanced MRI. Recent research has revealed that GM abnormalities manifest early in the course of the disease and predict both conversion to MS and the progressive accrual of disability [86]. Moreover, GM atrophy is more severe than WM atrophy in the early stages of the disease [87]. Total brain volume measurements hold significant clinical importance in MS diagnosis and monitoring. However, accurately measuring brain atrophy is crucial for detecting changes over short periods of time, and this is challenging in MS patients

compared to healthy individuals due to smaller brain volumes [88–90]. While volumetrics and their derived measurements have shown promise as prognosis biomarkers for MS, the estimation of total brain atrophy in MS patients is challenging and can only be achieved after several years of longitudinal follow-up [91]. Therefore, utilizing brain atrophy as a prognosis biomarker at an individual level, particularly in the early stages of the disease, is difficult [92–95]. Furthermore, spinal cord volumetrics, especially in the cervical segment, have been found to exhibit higher atrophy in MS patients than healthy controls. Additionally, the atrophy rate of the spinal cord is higher than that of the total brain, and patients with PPMS experience more atrophy than those with RRMS [96]. Cortical lesions have been shown to have better prognostic value for clinical outcomes and disability progression than WMLs. Therefore, further research on the diagnosis, monitoring, and treatment of MS should consider cortical lesions as a valuable target [97–99].

## Conclusion

This review provides evidence for the predictive potential of various MRI landmarks in MS. Lesions and changes within CNS structures such as white matter, gray matter, corpus callosum, thalamus, and spinal cord serve as potential indicators for predicting the progression of disability. Various prognostic factors are linked to the progression of MS, encompassing the presence of cortical lesions, alterations in gray matter volume, whole brain atrophy, the corpus callosum index, changes in thalamic volume, and lesions or modifications in the cross-sectional area of the spinal cord. Regarding cognitive impairment in individuals with MS, dependable predictors include cortical gray matter volume, brain atrophy, characteristics of lesions (such as T2-lesion load, temporal, frontal, and cerebellar lesions, volume of white matter lesions), thalamic volume, and density of the corpus callosum. Overall, MRI appears to be a useful tool for predicting MS disability progression, progression of disease, and cognitive decline.

### Limitations and suggestions

MRI has been widely used to detect and monitor MS related abnormalities. However, its limitations in predicting disease progression have been noted. Firstly, conventional MRI measures may lack specificity in predicting disease progression. The lesions seen on MRI may not always correlate with clinical symptoms or disease progression. Secondly, conventional MRI may not detect early pathological changes in MS, especially in the absence of visible lesions. This can limit its ability to predict disease progression accurately. MS is a complex disease with various underlying mechanisms such as inflammation, neurodegeneration, and remyelination that conventional MRI may not capture entirely, limiting its predictive value. Lastly, conventional MRI primarily focuses on structural changes and may not fully reflect functional impairment or disability progression in MS patients. Therefore, while MRI is useful in detecting MS-related abnormalities, its limitations in predicting disease progression should be taken into consideration. There are several ways to address these limitations. Advanced MRI techniques, including Diffusion Tensor Imaging (DTI), Magnetization Transfer Imaging (MTI), and Functional Magnetic Resonance Imaging (fMRI), offer precise measures of MS progression by capturing microstructural changes, myelin content, and functional connectivity alterations. Moreover, quantitative MRI measures, such as brain atrophy rates, lesion volumes, and magnetization transfer ratios, provide objective biomarkers when combined with clinical data. Additionally, long-term follow-up studies with repeated MRI scans and clinical assessments can identify imaging biomarkers that better correlate with disease progression. Furthermore, combining conventional MRI with other modalities like Positron Emission Tomography

(PET) or Optical Coherence Tomography (OCT) can offer a more comprehensive assessment of MS pathology and improve predictive accuracy.

## Supporting information

**S1 Checklist.**
(DOCX)

## Acknowledgments

The authors would like to thank the Clinical Research Development Unit of Poursina Hospital, Guilan University of Medical Sciences, Guilan, Iran.

## Author Contributions

**Conceptualization:** Nima Broomand Lomer, Alia Saberi.

**Methodology:** Nima Broomand Lomer, Alia Saberi.

**Supervision:** Alia Saberi.

**Writing – original draft:** Nima Broomand Lomer, Kamal AmirAshjei Asalemi, Alia Saberi, Kasra Sarlak.

**Writing – review & editing:** Nima Broomand Lomer, Kamal AmirAshjei Asalemi, Alia Saberi, Kasra Sarlak.

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
