## [Decision Letter · Decision Letter 0]

17 Jan 2024

PONE-D-23-33712Predictors of Multiple Sclerosis Progression: A Systematic Review of Conventional Magnetic Resonance Imaging StudiesPLOS ONE

Dear Dr. Broomand,

Thank you for submitting your manuscript to PLOS ONE. After careful consideration, we feel that it has merit but does not fully meet PLOS ONE’s publication criteria as it currently stands. Therefore, we invite you to submit a revised version of the manuscript that addresses the points raised during the review process.

REVIEWER 1: This is a review of the literature regarding the predictors of progression in multiple sclerosis (MS) patients. The authors made wide selection of the sources and found some interesting observations. However some of the results and conclusions are imprecise: the expressions like "some features of MS lesions" , "some MS lesion characteristics" or "some spinal cord markers"should be described much more precisely.Besides, the authors did not analyze the references regarding the use of the advanced MR techniques in MS, like: MR spectroscopy, MR diffusion, MR tensor diffusion, functional MRI), they just mentioned the advanced MR techniques n the discusssion section. This should be added.REVIEWER 2: Summary:This manuscript provides a review of 53 research works regarding the progression of multiple sclerosis. It defines three aspects of progression, i.e. progression of disease, progression of disability and cognitive decline, and then summarizes the MRI correlates.Comments:The review seems to miss important recent works. See a few examples below and the references therein:Branco, Dario, Beniamino di Martino, Antonio Esposito, Gioacchino Tedeschi, Simona Bonavita, and Luigi Lavorgna. “Machine Learning Techniques for Prediction of Multiple Sclerosis Progression.” Soft Computing 26, no. 22 (November 1, 2022): 12041–55. https://doi.org/10.1007/s00500-022-07503-z.Taloni, Alessandro, Francis Allen Farrelly, Giuseppe Pontillo, Nikolaos Petsas, Costanza Giannì, Serena Ruggieri, Maria Petracca, et al. “Evaluation of Disability Progression in Multiple Sclerosis via Magnetic-Resonance-Based Deep Learning Techniques.” International Journal of Molecular Sciences 23, no. 18 (September 13, 2022): 10651. https://doi.org/10.3390/ijms231810651.Storelli, Loredana, Matteo Azzimonti, Mor Gueye, Carmen Vizzino, Paolo Preziosa, Gioachino Tedeschi, Nicola De Stefano, Patrizia Pantano, Massimo Filippi, and Maria A. Rocca. “A Deep Learning Approach to Predicting Disease Progression in Multiple Sclerosis Using Magnetic Resonance Imaging.” Investigative Radiology 57, no. 7 (July 1, 2022): 423–32. https://doi.org/10.1097/RLI.0000000000000854.Pinto, Mauro F., Hugo Oliveira, Sónia Batista, Luís Cruz, Mafalda Pinto, Inês Correia, Pedro Martins, and César Teixeira. “Prediction of Disease Progression and Outcomes in Multiple Sclerosis with Machine Learning.” Scientific Reports 10, no. 1 (December 3, 2020): 21038. https://doi.org/10.1038/s41598-020-78212-6.My general impression of this survey is that it provides a "first order analysis" of the found literature, meaning that it simply lists certain aspects and results of the surveyed papers, lacking deeper comprehension of the subject.In the research works, this definition of "disability progression" may vary substantially. The review should elaborate on the definitions and identify the most common or robust one.Similar holds for "cognitive decline"; how is it defined? Since there are many cognitive tests and combinations thereof, and many aspects of cognition that can be tested; hence, the definition of cognitive decline is unlikely universal? Please elaborate on the definition and variations.It seems that the present survey does not consistently summarize or pool the results of the studies regarding the three outcome aspects (disability&disease progression and cognitive decline). Namely, certain results are reported in Table 1 in column "Correlations with MRI markers", however, the authors mostly reported the variables nad their interactions studied, but did not report the numerical evaluation and/or significance.Related to above, a table mapping the MRI measurements to the three aspects of progression, with indication of the association or even a numerical results would provide a clearer picture.It would be beneficial to understand from reading the review in which findings the study agree and in which they do not. This would provide an answer which findings could be generalized and which need further investigation. As it currently stands, the information seems to be partially extracted and reported, leaving it up to the reading to mentally comprehend this information and extract relevant findings, which otherwise should be the purpose of the survey.Please elaborate on the experimental design and statistical evaluation in the reviewed studies, indicating good practices to be followed by researches reading the review.Besides addressing the Reviewers comments and suggestions, please do brief on Benign and Malignant classifications of MS and their predictions of MS progression using MRI.REVIEWER 1: This is a review of the literature regarding the predictors of progression in multiple sclerosis (MS) patients. The authors made wide selection of the sources and found some interesting observations. However some of the results and conclusions are imprecise: the expressions like "some features of MS lesions" , "some MS lesion characteristics" or "some spinal cord markers"should be described much more precisely.Besides, the authors did not analyze the references regarding the use of the advanced MR techniques in MS, like: MR spectroscopy, MR diffusion, MR tensor diffusion, functional MRI), they just mentioned the advanced MR techniques n the discusssion section. This should be added.REVIEWER 2: Summary:This manuscript provides a review of 53 research works regarding the progression of multiple sclerosis. It defines three aspects of progression, i.e. progression of disease, progression of disability and cognitive decline, and then summarizes the MRI correlates.Comments:The review seems to miss important recent works. See a few examples below and the references therein:Branco, Dario, Beniamino di Martino, Antonio Esposito, Gioacchino Tedeschi, Simona Bonavita, and Luigi Lavorgna. “Machine Learning Techniques for Prediction of Multiple Sclerosis Progression.” Soft Computing 26, no. 22 (November 1, 2022): 12041–55. https://doi.org/10.1007/s00500-022-07503-z.Taloni, Alessandro, Francis Allen Farrelly, Giuseppe Pontillo, Nikolaos Petsas, Costanza Giannì, Serena Ruggieri, Maria Petracca, et al. “Evaluation of Disability Progression in Multiple Sclerosis via Magnetic-Resonance-Based Deep Learning Techniques.” International Journal of Molecular Sciences 23, no. 18 (September 13, 2022): 10651. https://doi.org/10.3390/ijms231810651.Storelli, Loredana, Matteo Azzimonti, Mor Gueye, Carmen Vizzino, Paolo Preziosa, Gioachino Tedeschi, Nicola De Stefano, Patrizia Pantano, Massimo Filippi, and Maria A. Rocca. “A Deep Learning Approach to Predicting Disease Progression in Multiple Sclerosis Using Magnetic Resonance Imaging.” Investigative Radiology 57, no. 7 (July 1, 2022): 423–32. https://doi.org/10.1097/RLI.0000000000000854.Pinto, Mauro F., Hugo Oliveira, Sónia Batista, Luís Cruz, Mafalda Pinto, Inês Correia, Pedro Martins, and César Teixeira. “Prediction of Disease Progression and Outcomes in Multiple Sclerosis with Machine Learning.” Scientific Reports 10, no. 1 (December 3, 2020): 21038. https://doi.org/10.1038/s41598-020-78212-6.My general impression of this survey is that it provides a "first order analysis" of the found literature, meaning that it simply lists certain aspects and results of the surveyed papers, lacking deeper comprehension of the subject.In the research works, this definition of "disability progression" may vary substantially. The review should elaborate on the definitions and identify the most common or robust one.Similar holds for "cognitive decline"; how is it defined? Since there are many cognitive tests and combinations thereof, and many aspects of cognition that can be tested; hence, the definition of cognitive decline is unlikely universal? Please elaborate on the definition and variations.It seems that the present survey does not consistently summarize or pool the results of the studies regarding the three outcome aspects (disability&disease progression and cognitive decline). Namely, certain results are reported in Table 1 in column "Correlations with MRI markers", however, the authors mostly reported the variables nad their interactions studied, but did not report the numerical evaluation and/or significance.Related to above, a table mapping the MRI measurements to the three aspects of progression, with indication of the association or even a numerical results would provide a clearer picture.It would be beneficial to understand from reading the review in which findings the study agree and in which they do not. This would provide an answer which findings could be generalized and which need further investigation. As it currently stands, the information seems to be partially extracted and reported, leaving it up to the reading to mentally comprehend this information and extract relevant findings, which otherwise should be the purpose of the survey.Please elaborate on the experimental design and statistical evaluation in the reviewed studies, indicating good practices to be followed by researches reading the review.Besides addressing the Reviewers comments and suggestions, please do brief on Benign and Malignant classifications of MS and their predictions of MS progression using MRI. ==============================

We look forward to receiving your revised manuscript.

Kind regards,

Asokan Govindaraj Vaithinathan

Academic Editor

PLOS ONE

Additional Editor Comments:

• REVIEWER 1: This is a review of the literature regarding the predictors of progression in multiple sclerosis (MS) patients. The authors made wide selection of the sources and found some interesting observations. However some of the results and conclusions are imprecise: the expressions like "some features of MS lesions" , "some MS lesion characteristics" or "some spinal cord markers"should be described much more precisely.

Besides, the authors did not analyze the references regarding the use of the advanced MR techniques in MS, like: MR spectroscopy, MR diffusion, MR tensor diffusion, functional MRI), they just mentioned the advanced MR techniques n the discusssion section. This should be added.

• REVIEWER 2: Summary:

This manuscript provides a review of 53 research works regarding the progression of multiple sclerosis. It defines three aspects of progression, i.e. progression of disease, progression of disability and cognitive decline, and then summarizes the MRI correlates.

Comments:

The review seems to miss important recent works. See a few examples below and the references therein:

Branco, Dario, Beniamino di Martino, Antonio Esposito, Gioacchino Tedeschi, Simona Bonavita, and Luigi Lavorgna. “Machine Learning Techniques for Prediction of Multiple Sclerosis Progression.” Soft Computing 26, no. 22 (November 1, 2022): 12041–55. https://doi.org/10.1007/s00500-022-07503-z.

Taloni, Alessandro, Francis Allen Farrelly, Giuseppe Pontillo, Nikolaos Petsas, Costanza Giannì, Serena Ruggieri, Maria Petracca, et al. “Evaluation of Disability Progression in Multiple Sclerosis via Magnetic-Resonance-Based Deep Learning Techniques.” International Journal of Molecular Sciences 23, no. 18 (September 13, 2022): 10651. https://doi.org/10.3390/ijms231810651.

Storelli, Loredana, Matteo Azzimonti, Mor Gueye, Carmen Vizzino, Paolo Preziosa, Gioachino Tedeschi, Nicola De Stefano, Patrizia Pantano, Massimo Filippi, and Maria A. Rocca. “A Deep Learning Approach to Predicting Disease Progression in Multiple Sclerosis Using Magnetic Resonance Imaging.” Investigative Radiology 57, no. 7 (July 1, 2022): 423–32. https://doi.org/10.1097/RLI.0000000000000854.

Pinto, Mauro F., Hugo Oliveira, Sónia Batista, Luís Cruz, Mafalda Pinto, Inês Correia, Pedro Martins, and César Teixeira. “Prediction of Disease Progression and Outcomes in Multiple Sclerosis with Machine Learning.” Scientific Reports 10, no. 1 (December 3, 2020): 21038. https://doi.org/10.1038/s41598-020-78212-6.

My general impression of this survey is that it provides a "first order analysis" of the found literature, meaning that it simply lists certain aspects and results of the surveyed papers, lacking deeper comprehension of the subject.

In the research works, this definition of "disability progression" may vary substantially. The review should elaborate on the definitions and identify the most common or robust one.

Similar holds for "cognitive decline"; how is it defined? Since there are many cognitive tests and combinations thereof, and many aspects of cognition that can be tested; hence, the definition of cognitive decline is unlikely universal? Please elaborate on the definition and variations.

It seems that the present survey does not consistently summarize or pool the results of the studies regarding the three outcome aspects (disability&disease progression and cognitive decline). Namely, certain results are reported in Table 1 in column "Correlations with MRI markers", however, the authors mostly reported the variables nad their interactions studied, but did not report the numerical evaluation and/or significance.

Related to above, a table mapping the MRI measurements to the three aspects of progression, with indication of the association or even a numerical results would provide a clearer picture.

It would be beneficial to understand from reading the review in which findings the study agree and in which they do not. This would provide an answer which findings could be generalized and which need further investigation. As it currently stands, the information seems to be partially extracted and reported, leaving it up to the reading to mentally comprehend this information and extract relevant findings, which otherwise should be the purpose of the survey.

Please elaborate on the experimental design and statistical evaluation in the reviewed studies, indicating good practices to be followed by researches reading the review.

*Besides the above comments of the Reviewers, it is suggested to explain the Benign and Malignant classifications of MS, does Benign and Malignant forms of classifications have any predictability on MS progression.

Reviewers' comments:

Reviewer's Responses to Questions

**Comments to the Author**

1. Is the manuscript technically sound, and do the data support the conclusions?

Reviewer #1: Partly

Reviewer #2: Partly

2. Has the statistical analysis been performed appropriately and rigorously? 

Reviewer #1: N/A

Reviewer #2: No

3. Have the authors made all data underlying the findings in their manuscript fully available?

Reviewer #1: Yes

Reviewer #2: No

4. Is the manuscript presented in an intelligible fashion and written in standard English?

Reviewer #1: Yes

Reviewer #2: Yes

5. Review Comments to the Author

Reviewer #1: This is a review of the literature regarding the predictors of progression in multiple sclerosis (MS) patients. The authors made wide selection of the sources and found some interesting observations. However some of the results and conclusions are imprecise: the expressions like "some features of MS lesions" , "some MS lesion characteristics" or "some spinal cord markers"should be described much more precisely.

Besides, the authors did not analyze the references regarding the use of the advanced MR techniques in MS, like: MR spectroscopy, MR diffusion, MR tensor diffusion, functional MRI), they just mentioned the advanced MR techniques n the discusssion section. This should be added.

Reviewer #2: Summary:

This manuscript provides a review of 53 research works regarding the progression of multiple sclerosis. It defines three aspects of progression, i.e. progression of disease, progression of disability and cognitive decline, and then summarizes the MRI correlates.

Comments:

The review seems to miss important recent works. See a few examples below and the references therein:

Branco, Dario, Beniamino di Martino, Antonio Esposito, Gioacchino Tedeschi, Simona Bonavita, and Luigi Lavorgna. “Machine Learning Techniques for Prediction of Multiple Sclerosis Progression.” Soft Computing 26, no. 22 (November 1, 2022): 12041–55. https://doi.org/10.1007/s00500-022-07503-z.

Taloni, Alessandro, Francis Allen Farrelly, Giuseppe Pontillo, Nikolaos Petsas, Costanza Giannì, Serena Ruggieri, Maria Petracca, et al. “Evaluation of Disability Progression in Multiple Sclerosis via Magnetic-Resonance-Based Deep Learning Techniques.” International Journal of Molecular Sciences 23, no. 18 (September 13, 2022): 10651. https://doi.org/10.3390/ijms231810651.

Storelli, Loredana, Matteo Azzimonti, Mor Gueye, Carmen Vizzino, Paolo Preziosa, Gioachino Tedeschi, Nicola De Stefano, Patrizia Pantano, Massimo Filippi, and Maria A. Rocca. “A Deep Learning Approach to Predicting Disease Progression in Multiple Sclerosis Using Magnetic Resonance Imaging.” Investigative Radiology 57, no. 7 (July 1, 2022): 423–32. https://doi.org/10.1097/RLI.0000000000000854.

Pinto, Mauro F., Hugo Oliveira, Sónia Batista, Luís Cruz, Mafalda Pinto, Inês Correia, Pedro Martins, and César Teixeira. “Prediction of Disease Progression and Outcomes in Multiple Sclerosis with Machine Learning.” Scientific Reports 10, no. 1 (December 3, 2020): 21038. https://doi.org/10.1038/s41598-020-78212-6.

My general impression of this survey is that it provides a "first order analysis" of the found literature, meaning that it simply lists certain aspects and results of the surveyed papers, lacking deeper comprehension of the subject.

In the research works, this definition of "disability progression" may vary substantially. The review should elaborate on the definitions and identify the most common or robust one.

Similar holds for "cognitive decline"; how is it defined? Since there are many cognitive tests and combinations thereof, and many aspects of cognition that can be tested; hence, the definition of cognitive decline is unlikely universal? Please elaborate on the definition and variations.

It seems that the present survey does not consistently summarize or pool the results of the studies regarding the three outcome aspects (disability&disease progression and cognitive decline). Namely, certain results are reported in Table 1 in column "Correlations with MRI markers", however, the authors mostly reported the variables nad their interactions studied, but did not report the numerical evaluation and/or significance.

Related to above, a table mapping the MRI measurements to the three aspects of progression, with indication of the association or even a numerical results would provide a clearer picture.

It would be beneficial to understand from reading the review in which findings the study agree and in which they do not. This would provide an answer which findings could be generalized and which need further investigation. As it currently stands, the information seems to be partially extracted and reported, leaving it up to the reading to mentally comprehend this information and extract relevant findings, which otherwise should be the purpose of the survey.

Please elaborate on the experimental design and statistical evaluation in the reviewed studies, indicating good practices to be followed by researches reading the review.

6. PLOS authors have the option to publish the peer review history of their article (what does this mean?). If published, this will include your full peer review and any attached files.

Reviewer #1: No

Reviewer #2: **Yes: **Žiga Špiclin

---

## [Author Response · Author response to Decision Letter 0]

25 Feb 2024

Response to Reviewer 1 Comments

This is a review of the literature regarding the predictors of progression in multiple sclerosis (MS) patients. The authors made wide selection of the sources and found some interesting observations. However, some of the results and conclusions are imprecise: 

Point 1: The expressions like "some features of MS lesions”, "some MS lesion characteristics" or "some spinal cord markers “should be described much more precisely.

Response 1: Thank you for your comment. We have revised the text to provide greater clarity and precision, addressing the issues raised in your thoughtful comment. Changes to these phrases can be found in (lines: 14-25, 281, 307-315)

Point 2: Besides, the authors did not analyze the references regarding the use of the advanced MR techniques in MS, like: MR spectroscopy, MR diffusion, MR tensor diffusion, functional MRI), they just mentioned the advanced MR techniques in the discussion section. This should be added.

Response 2: Thank you for your comment. The aim of this research work was to assess the applicability of conventional MRI in predicting the progression of MS, which is also presented in the inclusion criteria of this manuscript. However, our goal was not to analyze the articles concerning their data derived from advanced MRI techniques.

Response to Reviewer 2 Comments

This manuscript provides a review of 53 research works regarding the progression of multiple sclerosis. It defines three aspects of progression, i.e., progression of disease, progression of disability and cognitive decline, and then summarizes the MRI correlates. Comments:

Point 1: The review seems to miss important recent works. See a few examples below and the references therein:

Branco, Dario, Beniamino di Martino, Antonio Esposito, Gioacchino Tedeschi, Simona Bonavita, and Luigi Lavorgna. “Machine Learning Techniques for Prediction of Multiple Sclerosis Progression.” Soft Computing 26, no. 22 (November 1, 2022): 12041–55. https://doi.org/10.1007/s00500-022-07503-z.

Taloni, Alessandro, Francis Allen Farrelly, Giuseppe Pontillo, Nikolaos Petsas, Costanza Giannì, Serena Ruggieri, Maria Petracca, et al. “Evaluation of Disability Progression in Multiple Sclerosis via Magnetic-Resonance-Based Deep Learning Techniques.” International Journal of Molecular Sciences 23, no. 18 (September 13, 2022): 10651. https://doi.org/10.3390/ijms231810651.

Storelli, Loredana, Matteo Azzimonti, Mor Gueye, Carmen Vizzino, Paolo Preziosa, Gioachino Tedeschi, Nicola De Stefano, Patrizia Pantano, Massimo Filippi, and Maria A. Rocca. “A Deep Learning Approach to Predicting Disease Progression in Multiple Sclerosis Using Magnetic Resonance Imaging.” Investigative Radiology 57, no. 7 (July 1, 2022): 423–32. https://doi.org/10.1097/RLI.0000000000000854.

Pinto, Mauro F., Hugo Oliveira, Sónia Batista, Luís Cruz, Mafalda Pinto, Inês Correia, Pedro Martins, and César Teixeira. “Prediction of Disease Progression and Outcomes in Multiple Sclerosis with Machine Learning.” Scientific Reports 10, no. 1 (December 3, 2020): 21038. https://doi.org/10.1038/s41598-020-78212-6.

Response 1: Thank you for your thoughtful comment. The primary goal of our study was to evaluate the applicability of conventional MRI in predicting the progression of MS with classic human-based analysis. We excluded articles in which deep learning and machine learning models were used to predict outcomes. Apart from the fact that our study did not aim for this approach, incorporating studies utilizing AI would lead to the inclusion of numerous eligible studies, potentially complicating the readability and comprehension of our study for all readers.

Point 2: My general impression of this survey is that it provides a "first order analysis" of the found literature, meaning that it simply lists certain aspects and results of the surveyed papers, lacking deeper comprehension of the subject.

Response 2: Thank you for your comment. As mentioned in the study, we aimed to see which neuroimaging markers of conventional MRI can be used to predict the course of MS. We have identified and analyzed the key aspects of the literature that met our inclusion criteria. In the results and discussion section, we have categorized the changes in imaging features. We have also highlighted the significant alterations and proven findings that are consistent across various studies. These findings can be used by experts in the field for clinical decision-making and incorporated into guidelines that outline the anti-criteria of McDonald criteria for MS. As you know, the McDonald Criteria have undergone rapid evolution, with revisions occurring in 2005, 2010, and most recently in 2017. Each revision aims to simplify the criteria, uphold diagnostic precision, minimize misdiagnosis risks, and facilitate earlier MS detection. As research progresses and new diagnostic tools and biomarkers emerge, it is anticipated that the McDonald Criteria will continue to evolve to accommodate emerging clinical and imaging findings, alongside the evolution of new treatment guidelines.

Point 3: In the research works, this definition of "disability progression" may vary substantially. The review should elaborate on the definitions and identify the most common or robust one.

Response 3: Thank you for your comment. Depending on which disability assessment measure reported in each study, we employed three distinct methods to present disability outcomes: EDSS, 9HPT, and T25FWT. An elevation in any of these measures indicates higher disability levels. Definitions of each measure are included in the methods section between lines 110-120.

Point 4: Similar holds for "cognitive decline"; how is it defined? Since there are many cognitive tests and combinations thereof, and many aspects of cognition that can be tested; hence, the definition of cognitive decline is unlikely universal? Please elaborate on the definition and variations.

Response 4: Thank you for your comment. Indeed, cognitive function assessment employs various tools. Among them, the SDMT and PASAT are prominently used, with their descriptions provided in the methods section, lines 126-130. Other studies employ different specific tests for this purpose. Narrowing down to only one or two primary tools for cognitive function evaluation would necessitate excluding many important studies with no reasonable logic. Additional less frequent tests utilized in these studies include: 

MMSE (Mini-Mental State Examination)

RCPM (Raven’s Colored Progressive Matrices)

RBMT (Rivermead Behavioral Memory Test)

SPS (Standardized profile score)

SS (Screening scores)

TMT (Trail Making Test)

WF (Word fluency)

BVMT (Brief Visuospatial Memory Test)

RAVLT (Rey Auditory Verbal Learning Test)

DSST (Digit Symbol Substitution Test)

Point 5: It seems that the present survey does not consistently summarize or pool the results of the studies regarding the three outcome aspects (disability & disease progression and cognitive decline). Namely, certain results are reported in Table 1 in column "Correlations with MRI markers", however, the authors mostly reported the variables and their interactions studied, but did not report the numerical evaluation and/or significance.

Response 5: Thank you for your comment. We have updated the table along with the relevant statistical and numerical measurements for each section. However, initially, we had included all statistical measures for each section, but due to the table's high volume, we had to remove them.

Point 6: Related to above, a table mapping the MRI measurements to the three aspects of progression, with indication of the association or even a numerical result would provide a clearer picture.

Response 6: Thank you for your comment. We have updated the table along with the relevant statistical and numerical measurements for each section. However, initially, we had included all statistical measures for each section, but due to the table's high volume, we had to remove them.

Point 7: It would be beneficial to understand from reading the review in which findings the study agrees and in which they do not. This would provide an answer which findings could be generalized and which need further investigation. As it currently stands, the information seems to be partially extracted and reported, leaving it up to the reading to mentally comprehend this information and extract relevant findings, which otherwise should be the purpose of the survey.

Response 7: Thank you for your comment. As previously stated, we've pinpointed and examined the crucial elements of the literature that align with our inclusion criteria. Within the results and discussion section, we've organized shifts in imaging characteristics, emphasizing notable changes and established findings that are consistently observed across multiple studies. These insights can aid clinicians in making informed decisions and may contribute to the revision of guidelines detailing the non-criteria of the McDonald criteria for MS.

Point 8: Please elaborate on the experimental design and statistical evaluation in the reviewed studies, indicating good practices to be followed by researches reading the review.

Response 8: Thank you for your comment. We added the design of study section for every included study in the table in “Study” section after year of the study. “C/S” and “C” represent cross-sectional and cohort study type, respectively. Here again, initially, we had included all study designs for included studies but due to the table's high volume, we had to remove them. As this article is intended for use by clinicians, including both statistical and other types of data will not be helpful for those who want to use it in a clinical setting. In fact, it may only cause confusion and make the article feel cluttered and unhelpful.

Response to Academic Editor’s Comments

Point 1: Besides addressing the Reviewers comments and suggestions, please do brief on Benign and Malignant classifications of MS and their predictions of MS progression using MRI.

Response 1: Thank you for your comment. Our study aimed to go beyond simply categorizing studies into benign (EDSS ≤3.0 after 15 years of disease duration) or malignant (EDSS ≥6 within 5 years from symptom onset) forms. Rather, our objective was to examine the changes in MS over time across these three dimensions considering the inclusion criteria stated in the manuscript. Specifically, our focus was on evaluating the imaging correlations of MS in these three aspects with the intention of informing clinical decision-making and treatment selection. Additionally, we sought to identify imaging features that could potentially aid in preventing progression to the progressive form of the disease according to the inclusion criteria.

---

## [Editor Report · Decision Letter 1]

27 Feb 2024

Predictors of Multiple Sclerosis Progression: A Systematic Review of Conventional Magnetic Resonance Imaging Studies

PONE-D-23-33712R1

Dear Dr.Nima Broomand,

We’re pleased to inform you that your manuscript has been judged scientifically suitable for publication and will be formally accepted for publication once it meets all outstanding technical requirements.

Kind regards,

Asokan Govindaraj Vaithinathan

Academic Editor

PLOS ONE
---

## [Editor Report · Acceptance letter]

5 Mar 2024

PONE-D-23-33712R1 

PLOS ONE

Dear Dr. Broomand Lomer, 

I'm pleased to inform you that your manuscript has been deemed suitable for publication in PLOS ONE. Congratulations! Your manuscript is now being handed over to our production team.

Kind regards, 

on behalf of

Dr. Asokan Govindaraj Vaithinathan 

Academic Editor

PLOS ONE